# Quantifying climate model representation of the wintertime Euro-Atlantic circulation using geopotential-jet regimes

Joshua Dorrington[1,*], Kristian Strommen[1,*], and Federico Fabiano[2]

[1]Department of Atmospheric, Oceanic, and Planetary Physics, University of Oxford, UK
[2]Institute of Atmospheric Sciences and Climate (ISAC-CNR), Bologna, Italy
[*]These authors contributed equally to this work

**Correspondence:** Joshua Dorrington (joshua.dorrington@physics.ox.ac.uk)

**Abstract.** Even the most advanced climate models struggle to reproduce the observed wintertime circulation of the atmosphere over the North Atlantic and Western Europe. During winter, the large-scale motions of this particularly challenging region are dominated by eddy-driven and highly non-linear flows, whose low frequency variability is often studied from the perspective of regimes – a small number of qualitatively distinct atmospheric states. Poor representation of regimes associated with persistent atmospheric blocking events, or variations in jet latitude, degrade the ability of models to correctly simulate extreme events. In this paper we leverage a recently developed hybrid approach – which combines both jet and geopotential height data – to assess the representation of regimes in 8,400 years of historical climate simulations drawn from CMIP6, CMIP5 and HighResMip. We show that these geopotential-jet regimes are particularly suited to the analysis of climate data, with considerable reductions in sampling variability compared to classical regime approaches. We find that CMIP6 has a considerably improved spatial regime structure, and a more trimodal eddy-driven jet, relative to CMIP5, but still struggles with underpersistent regimes and too little European blocking when compared to reanalysis. Reduced regime persistence can be understood, at least in part, as a result of jets that are too fast and eddy feedbacks on the jet stream that are too weak - structural errors that do not noticeably improve in higher resolution models.

## 1 Introduction

### 1.1 Motivation

There are very few regions of the atmosphere which prove so stubbornly difficult to model as the Euro-Atlantic troposphere. Accurately predicting its evolution, or even modelling its climate, requires correctly capturing the interactions of breaking Rossby waves with orography and with a meandering jet stream, the development and maintenance of persistent blocking events, and the myriad external forcings that drive the tropospheric flow: ocean heat fluxes (Delworth and Zeng, 2016; Delworth et al., 2017), Arctic sea ice (Barnes and Screen, 2015), stratospheric signals (Domeisen et al., 2020) and tropical teleconnections (Rodríguez-Fonseca et al., 2016; Jiménez-Esteve and Domeisen, 2018) to name a few. Boreal Wintertime (DJF) is particularly challenging in this regard due to the particularly prominent role persistent blocking (Barriopedro et al., 2006) and trimodal jet dynamics (Woollings et al., 2010) play during this season. The implications of this complexity are far-reaching. Within

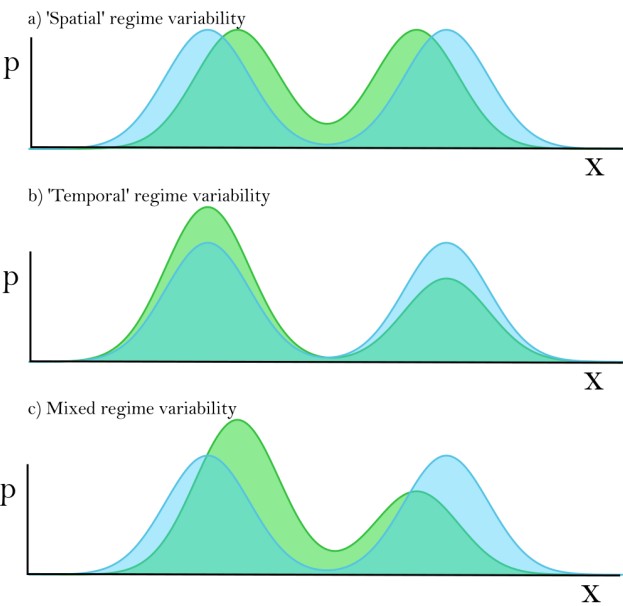

**Figure 1.** A schematic showing different kinds of regime variability, illustrated with a bimodal, 1D probability distribution, the two peaks of which could be considered as regimes. a) Spatial variability: the position of the two regime centroids is different between the two distributions, but their frequency of occurrence is unchanged. b) Temporal variability: The two regime centroids are located in the same position on the x-axis in both distributions, but the occurrence frequencies are different. c) Mixed variability: Here the regimes differ both temporally and spatially between the two distributions, showing shifts in the positions of the regimes, and their occurrence frequencies. In this case there is no unambiguous way to relate the regimes of one distribution to those of the other.

the context of weather forecasting, it results in poor seasonal predictability over Europe (Johnson et al., 2019) and within the context of climate modelling, it causes biases in model representations of historical Euro-Atlantic dynamics (Davini and D'Andrea, 2020) and a wide divergence in model predictions of the impacts of climate change on Europe, hindering the ability to develop long-term mitigation plans (Shepherd, 2014). If we are to improve our models of the Euro-Atlantic we must understand exactly what it is we are getting wrong in the first place. In such a strongly non-linear flow, evaluating simple bulk statistics, such as the mean zonal wind or mean pressure anomalies can be misleading, with the potential to obscure compensating or flow-dependent errors. The analysis of circulation regimes provides a valuable framework for studying such strongly nonlinear flows, allowing for an explicit consideration of large scale flow-dependence, with the added benefits of being computationally cheap and easy to diagnose from commonly archived variables.

While there is no agreed upon definition of a regime, all regime approaches aim to identify a small set of qualitatively distinct flow configurations, coarse-graining the continuously evolving atmospheric state into a series of transitions between this finite set of regimes. Such a regime model approximates the true variability of the system in two different ways. The first way is through the 'spatial' structure of the regimes – the different positions of the regime centroids in phase space (or similarly, the

different anomaly composites associated with the regimes in some target field of interest) which capture the large-scale weather patterns considered representative of the system. The second way is provided by the 'temporal' structure of the regimes, namely the occurrence frequency and lifetime distributions of each of those regimes. When modelling regime transitions as a Markovian process, as is frequently done, this temporal structure can be understood completely by fitting a transition matrix to the regime state sequence, which may be done as part of the clustering as in hidden Markov model approaches, or after the fact as in the K-means clustering approach. This reduces each lifetime distribution to a single day-to-day persistence probability. Differences in regimes – whether between two reanalysis time periods, between models and reanalysis, between two models, or between two model ensemble members – must manifest by projecting onto variations in the spatial or temporal structure of the regimes, or both. Figure 1 visualises these three possibilities, using a hypothetical bimodal, one-dimensional regime system as an illustrative example.

Regimes have a long history (extensively reviewed in Hannachi et al. (2017)), developing from the concept of recurrent weather types used in early operational forecasting (Bowie and Weightman, 1914; Gold, 1920; Baur et al., 1944; Rossby, 1940) and put on firmer theoretical ground with the seminal work of Charney and DeVore (1979) which proposed a link between persistent regimes and theoretically stable configurations of the large-scale flow. Euro-Atlantic weather regimes, identified via K-means clustering of 500 hPa geopotential height data (Z500), were introduced in Michelangeli et al. (1995), and have tremendous utility in helping to understand the flow dependent predictability of the atmosphere (Frame et al., 2013; Ferranti et al., 2015; Matsueda and Palmer, 2018), in developing a holistic picture of model biases (Dawson et al., 2012; Fabiano et al., 2020) and in modulating the impact of remote teleconnections, such as from the Madden-Julian Oscillation (Cassou, 2008) or the stratosphere (Charlton-Perez et al., 2018; Beerli and Grams, 2019). Approaches generalised to cover the entire year have also been used to understand flow dependent impacts in applied settings, such as energy generation (Grams et al., 2017; Van Der Wiel et al., 2019; Garrido-Perez et al., 2020). A complementary perspective, focusing on regime behaviour in the latitudinal meandering of the eddy-driven Atlantic jet stream, has also developed over the last decade (Woollings et al., 2010; Franzke et al., 2011; Madonna et al., 2017).

The goal of this paper is to perform a comprehensive assessment of how well state-of-the-art climate models represent historical regime behaviour in the wintertime Euro-Atlantic, and to what extent CMIP6 improves over CMIP5 in this regard. To achieve this, we will be making use of a novel framework, first developed in Dorrington and Strommen (2020) (hereafter DS20) which brings together the two main regime perspectives, – circulation regimes diagnosed from Z500 anomalies, and jet regimes diagnosed from the jet latitude index – in order to develop a more holistic understanding of mid-latitude variability. Expecting the reader to familiarise themselves with a new framework being applied to a huge amount of data warrants some justification, which we now aim to provide. The key motivation is the fact that most applications of regime analysis to weather forecasting and climate change implicitly rely on the assumption that regime variability, both across time and between models, is strictly temporal (as in Figure 1b), whereas classical and widely used methods based on clustering Z500 data produce regimes where the variability is very much mixed (as in Figure 1c).

To justify this assertion, we remind the reader of the main applications. Firstly, for weather forecasting, regimes are used to simplify European weather forecasts by focusing on the probability of transitions between regimes within a given forecast

window. Surface teleconnections associated with each regime can then be used to give a 'first-order' weather forecast, and the historical probability of certain regime transitions can give predictability higher than simple climatology at longer lead times. This widely used approach (Ferranti et al., 2015; Lavaysse et al., 2018; Matsueda and Palmer, 2018; Cortesi et al.,

2019) clearly breaks down if the regimes one can transition into at the time of forecast differ markedly from historical regimes, as neither climatological transition rates nor historical surface impacts can be expected to apply anymore. In Supplementary Figure 1, we show a 'worst-case' example of how the inclusion or not of a single year can notably change the regime patterns diagnosed using classical methods, showing that such a potential break-down cannot be easily ruled out. The second main application is to climate change. In the idealised Lorenz '63 regime system, Palmer (1999) noted that external forcing acted

to alter the occurrence rates of regimes (their temporal variability), but not their spatial structure: the regime patterns stay the same, but their persistence and occurrence rates change. This prompted Palmer to suggest the strategy of understanding future changes to Euro-Atlantic dynamics in terms of just a handful of numbers, namely the changes to persistence and occurrence rates of the regimes (see Corti et al. (1999); Cattiaux et al. (2013); Ullmann et al. (2014); Fabiano et al. (2021) for various attempts at implementing this strategy). As before, associated surface impacts can be estimated from this, giving a 'first-order'

approximation of climate change, and, as before, the strategy breaks down if the regime patterns change considerably in the future.

The presence of mixed variability is perhaps even more obviously problematic when evaluating model performance, whether by reference to observational data or other models. As an example, suppose one has diagnosed a regime pattern corresponding to the positive phase of the North Atlantic Oscillation (NAO), i.e. an NAO+ regime, and suppose that one is interested in

comparing the persistence of this regime in a given model with the regime persistence estimated using observational data. If the regime patterns are identical, this is easy, but in practice this is rarely true. Indeed, the pattern correlation between regimes in models and observations can easily be as low as 0.4 (Fabiano et al., 2020). For the NAO+ regime, which corresponds to a zonal jet, such low pattern correlations typically correspond to biases in the latitudinal position of the jet. Crucially, such mean state biases in the jet would be expected to produce changes in the persistence time-scales of the regime simply because

equatorwards-shifted jets are more persistent than poleward-shifted jets (Barnes and Hartmann, 2010). In particular, it becomes virtually impossible to infer if model biases in regime persistence are due to genuine deficiencies in the representation of physical processes, such as transient eddy feedbacks (Shutts, 1983), or are simply artefacts of having compared in-equivalent regimes. In general, it is unlikely that differences in the spatial pattern between any two regimes can be assumed to have a negligible impact on its persistence and occurrence statistics.

The fact that Z500 regime patterns are not generally comparable across different data sets is well known. This is even true when comparing ensemble members of an initialised forecast using a single model, prompting Falkena et al. (2021) to propose modifying the clustering algorithm in order to artificially force more consistent patterns. The failure of time-invariance across 20th century observational data was reported in DS20, which showed that standard regime diagnostics find notably different patterns depending on which time period one considers. In practice, many prior regime studies, both in the context of numerical

weather prediction (Ferranti et al., 2015) and climate modelling (Huth, 2000; Driouech et al., 2010; Ullmann et al., 2014; Fabiano et al., 2021), sidestep such issues by simply ignoring variability in the spatial regime structure. This is often done by

prescribing particular regime patterns as identified in a particular dataset of choice, such as a particular reanalysis product, and then identifying regime statistics in other datasets a posteriori. However, as the discussion in the preceding paragraph makes clear, this likely introduces additional uncertainties into analysis. Given the strong internal variability in the Euro-Atlantic circulation (Smith et al., 2019), this is clearly not ideal, and indeed there has arguably been little progress made in realising the strategy of Palmer (1999), let alone addressing the question of what determines if a model will have realistic regimes or not.

Of course, despite suggestive results using toy-models like Lorenz '63, there is no reason a priori to believe the Euro-Atlantic circulation does exhibit regimes that are not only time-invariant but also well captured by the majority of climate and forecast models. Mixed regime variability, time-varying patterns and the general difficulty of defining unambiguous regime patterns may simply be sources of uncertainty we have to live with. On the other hand, it is striking that much of this uncertainty is not present when defining regimes from the perspective of the eddy-driven jet latitude. While significant deviations from Gaussianity are hard to detect using Z500 data (Stephenson et al., 2004; Strommen et al., 2019), the daily distribution of jet latitudes is visibly trimodal, with peaks robustly centred at particular latitudes, and there is little to no variability in this structure across the 20th century (cf. Supplementary Figure 2). Guided by this puzzling mismatch between the two regime approaches, DS20 identified three Z500 regimes in the ERA20C reanalysis with vanishing inter-decadal variability in their spatial structure, obtained by filtering out the linear variability of the eddy-driven jet speed prior to clustering. DS20 posited that the reason the unambiguous multimodality of the jet latitude can't be seen in the the Z500 phase space is partly due to the confounding influence of the *longitudinal* variability of the jet, as characterised by its speed. Indeed, the jet speed is distributed unimodally, experiences considerable interdecadal variability (Woollings et al., 2014), and is largely decoupled from the jet latitude, with the two quantities uncorrelated on daily time scales (cf. Supplementary Figure 3).

Several studies have highlighted the different nature of the latitudinal and longitudinal variability of the jet. Besides its large decadal variability, the jet speed appears to be mostly unpredictable on seasonal time scales, unlike the jet latitude (Parker et al., 2019; Strommen, 2020). The two quantities also respond differently to thermal forcing (Baker et al., 2017), implying their response to climate change may be very different. As such, it is natural to want to treat these two quantities separately, and this approach has often been adopted in earlier studies (Woollings and Blackburn, 2012; Barnes and Polvani, 2013b). DS20 essentially argued that this separation should also be used in the context of regime analysis. This ethos, which is summarised in schematic Figure 2, will be the one adopted here.

## 1.2 Outline

In this paper, we first show that the highly stable spatial structure of the three regimes found in DS20 for ERA20C can be seen in an additional four reanalysis products, including the recent ERA5 reanalysis (Hersbach et al., 2020). By applying the same jet filtering methodology to model data, we also show that the same three regimes, which we here term *geopotential-jet regimes* due to their hybrid nature, can be found in the majority of coupled CMIP5, CMIP6 and HighResMip model simulations. In particular, the geopotential-jet regime patterns are to very good approximation time-invariant in both reanalysis data and model simulations, and the patterns seen in reanalysis are extremely well captured by the models. In other words, and in contrast to classical regimes, geopotential-jet regime variability is almost exclusively temporal. We view this as strong evidence for the

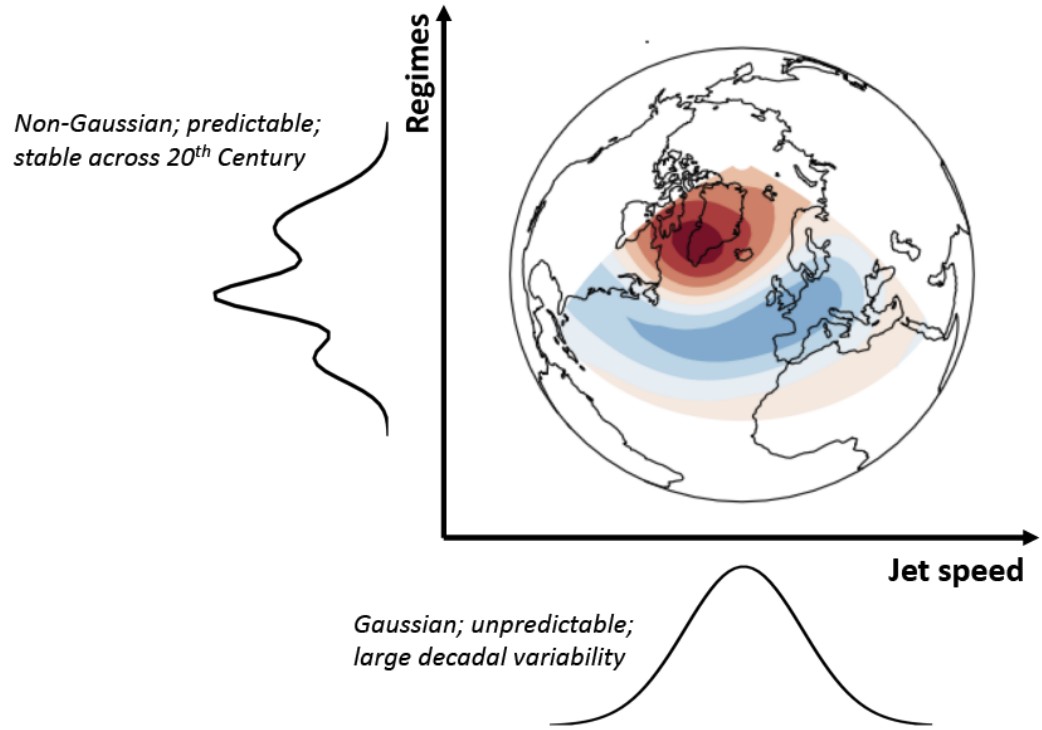

**Figure 2.** A schematic of the conceptual framework used in this paper. The Euro-Atlantic circulation, visualised here using the North Atlantic Oscillation (filled contours), is decomposed into orthogonal (i.e., uncorrelated) modes of jet variability: the longitudinal variability ('pulsing'), as measured by the Gaussian jet speed, and the latitudinal variability ('wobbling' or 'meandering'), as measured using non-Gaussian regimes. These two modes are then studied independently: the jet speed with linear methods (linear regression), and regimes with non-linear methods (changes to persistence and occurrence).

existence of time-invariant and model-reproducible regimes *a posteriori*. Because our jet filtering approach produces a phase space distribution which is unambiguously non-Gaussian and easily comparable to the trimodal jet structure, the reproducibility of geopotential-jet regimes across model simulations and different reanalysis can also be considered as strong evidence for the existence of Euro-Atlantic regimes full stop.

The stability (i.e. time-invariance) of geopotential-jet regimes in models, along with their fidelity (i.e. the close resemblance of their regime patterns to those in reanalysis), underpin the robustness of all further analysis performed. Firstly, we are able to easily pick out the minority of models with particularly poor spatial regime structure and can clearly relate such deficiencies to deficiencies in the representation of the trimodal jet, e.g. a failure to simulate trimodality. We show a clear reduction in the number of such 'bad' models from CMIP5 to CMIP6, primarily due to a reduction in the number of models with unimodal

jets. Next, we assess regime variability in models over the historical period 1900-2015, including biases in regime occurrence, persistence, and interdecadal variability of both. Finally, we ultimately want to understand why some models produce more

| Reanalysis name | Generating Centre | Year of production | Time period used | Observations included | Assimilating Model | Resolution (xy,z,t) |
|---|---|---|---|---|---|---|
| 20CRv2 | NOAA-CIRES | 2011 | 1900-2010 | surface observations | GFS (2008 experimental*) | (210km, 28 levels, 6 hrs) |
| ERA20C | ECMWF | 2016 | 1900-2010 | surface pressure and marine wind observations | IFS Cy38r1 | (125km, 91 levels, 3 hrs) |
| CERA20C | ECMWF | 2018 | 1901-2010 | surface, wind and ocean observations | IFS Cy41r2 | (125km, 91 levels, 3 hrs) |
| ERA5 | ECMWF | 2020 | 1950-2010 | surface, satellite and upper air observations | IFS Cy41r2 | (31km, 137 levels, 1hr) |
| 20CRv3 | NOAA-CIRES-DOE | 2021 | 1900-1980 | surface observations | GFS 14.01 | (60km, 64 levels, 6 hrs) |

**Table 1.** A summary of the 5 reanalysis products analysed in this paper. For references to full descriptions of the products, see main text. * 'Experimental' indicates that this was not an operational forecast model cycle

realistic regimes than others. As an initial investigation into this important question, we look at a number of features that might influence how well regimes are represented by models, including their horizontal resolution, strength of local eddy feedbacks, and the representation of remote teleconnection patterns known to influence the North Atlantic. A multilinear model is used
to show that much of the intermodel spread in regime structure can be predicted from these more general features, which thus provide useful guides for future research.

In section 2 we describe the model data and variables we analyse, provide a full description of the metrics and regime identification methodology we use, and describe the model features we have considered as possible explanatory variables for regime representation. Section 3 analyses how realistically and reliably climate models are able to reproduce regime patterns
found in reanalysis, and shows that the three geopotential-jet regimes found in DS20 are also optimal for the analysis of CMIP data. We also consider the relation between geopotential-jet regime and jet latitude regime representation. Section 4 looks at the occurrence and persistence of those regimes in models, and explores both mean-state biases and historical variability. Section 5 analyses the covariability in models' mean-state with regime statistics in order to gain insight into the causes of regime structure and model biases. Finally we summarise our key findings in section 6.

## 2  Data and methodology

### 2.1  Data

In order to assess the uncertainty in the historical record of regimes, we make use of 5 different reanalysis products. In the case of reanalyses with multiple equivalent ensemble members, we always use the first member. For each reanalysis we make use of boreal winter (DJF) geopotential height at 500 hPa (Z500) and zonal wind speed data at 850 hPa (U850), at a daily temporal
resolution and linearly interpolated onto a 1 degree grid, from ERA20C (Poli et al., 2016), the extended ERA5 (Hersbach et al.,

2020), CERA20C (Laloyaux et al., 2018), 20CRv2 (Compo et al., 2011), and 20CRv3 (Slivinski et al., 2021). Of these only CERA20C uses a coupled ocean-atmosphere model. Reanalyses produced by the same centre will share some similarities in the features of the assimilating model and in data-assimilation procedures, and so therefore are not totally independent. All available data covering the time period 1900-2010, available at the time of writing, was used. A summary of each product is given in Table 1.

We analyse equivalent model geopotential height and wind speed data drawn from the 5th (CMIP5) and 6th (CMIP6) phases of the coupled model inter-comparison project: multi-centre ensembles of earth-system models representing the state-of-the-art in global climate modelling in 2011 and 2020 respectively. We analyse the historical experiments for 31 single-member CMIP6 models (Eyring et al., 2016) detailed in SI table 1, and a total of 71 ensemble members from 28 distinct CMIP5 models (Taylor et al., 2012) detailed in SI table 2. These historical experiments consist of coupled uninitialised climate runs forced with historical greenhouse gas and aerosol forcings over the 20th century, after a spin-up from a free-running pre-industrial control run.

In section 5 we also make use of model data produced as part of the PRIMAVERA project, detailed in SI table 3. These coupled simulations all follow the HighResMIP protocol (Haarsma et al., 2016), and are therefore initialised in 1950, following a short 50-year spin-up. The simulations span the 65 years between 1950 and 2015, and use the same historical forcings as CMIP6. Six underlying models were used, each run at a number of different resolutions: CMCC-CM2 (Cherchi et al., 2019), CNRM-CM6 (Voldoire et al., 2019), EC-Earth3 (Haarsma et al., 2020), ECMWF-IFS (Roberts et al., 2018), HadGEM3-GC31 (Williams et al., 2018), MPI-ESM1-2 (Gutjahr et al., 2019), and AWI-CM-1.0 (Sein et al., 2017).

In total, we consider more than 8400 DJF seasons of climate model data, deriving from 128 ensemble integrations of 76 different climate models, originating from 18 independent modelling centres. We are therefore able to provide a comprehensive analysis of the current state-of-the-art in climate model regime representation.

## 2.2 Jet stream metrics

The structure of the low-level eddy-driven Atlantic jet can be summarised through the jet latitude and jet speed indices introduced in Woollings et al. (2010), which we compute with the simplified methodology of Parker et al. (2019). In brief, the jet speed index is defined as the maximum (oriented Eastward) of latitudinally averaged 850 hPa zonal wind speed, smoothed to remove synoptic fluctuations, over the Atlantic domain [0–60W,15-75N]. The jet latitude index is complementarily defined as the latitude at which the jet speed is maximum on a given day. The smoothing timescale varies in the literature, but here we apply a 5-day lowpass filter to the winds. Note that this methodology defines a unique latitude and speed for every day, and so cannot account for split jets or insignificant jet activity, and also does not contain any information on the meridional tilt of the jet.

As the probability distribution of the jet latitude in reanalysis is trimodal, it is difficult to summarise deficiencies in model jet latitude distribution through simple summary statistics such as the mean or variance. In order to holistically quantify the error in the jet latitude and jet speed distributions in models, we make use of the Wasserstein distance. Also known as the "Earth-mover distance", this metric provides a natural way to compare two probability distributions, and can be understood

informally as a measure of how much of the probability density of one distribution must be shifted to transport it into another. Formally, the Wasserstein distance (as used here) is defined as the integrated difference between two distributions' cumulative density functions, $\mu(x)$ and $\nu(x)$:

$$\text{WD}(\mu, \nu) = \int\limits_{-\infty}^{\infty} |\nu(x) - \mu(x)| dx \tag{1}$$

This distance was introduced to the atmospheric literature in Ghil (2015) and has since been applied to the analysis of both simple climate attractors (Robin et al., 2017) and fully coupled climate models (Vissio et al., 2020), the latter of which uses it to evaluate model error, just as we do here. Just as for the more common Euclidean distance, values close to zero indicate smaller differences.

Finally, a simpler measure of how well a model captures the jet latitude distribution can be given by identifying how many peaks a given model distribution has (i.e., whether it is trimodal, bimodal or unimodal) and the location of these peaks. To do this objectively, we used the python algorithm scipy.signal.find_peaks to locate, for a given distribution, all peaks with a height of at least 0.01 which are separated from each other by at least 4 degrees. The numbers were chosen based on inspecting the distribution for ERA5 and the CMIP6 multimodel mean. Note that this methodology excludes 'shoulders' from being classified as peaks.

### 2.3 Regime computation

We identify atmospheric regimes using the well-established methodology (Michelangeli et al., 1995; Cassou, 2008; Dawson et al., 2012) of computing the leading principal components of the daily mean 500hPa geopotential height field (Z500), restricted to Boreal winter (DJF) and the Euro-Atlantic region [30-90N,80W-40E], and partitioning this low-dimensional space into regimes using the K-means clustering algorithm, either with or without a prefiltering step. Principal components are calculated using the *eofs* Python package (Dawson, 2016). We refer to the standard approach, which is to directly cluster a number of the leading principal components, without any prefiltering, as producing *Classical Circulation Regimes*.

As discussed in the introduction, DS20 argued that robust identification of non-linear regime structure in the Z500 phase space is confounded by impact of noisy and predominately linear jet speed variability. Following the approach they introduced, we consider decomposing each of the principal components of Z500 into a component linearly related to jet speed variations, and a residual component, which we hypothesise to capture the bulk of the nonlinear variability:

$$\text{PC}_n(t) = A_n * u_{\text{jet}}(t) + \text{PC}_{n,\text{resid}}(t) + c_n \tag{2}$$

Where $\text{PC}_n(t)$ is the nth principal component of Z500, $u_{\text{jet}}(t)$ is jet speed, and $A_n$ and $c_n$ are a slope and intercept obtained by a linear best fit. We then cluster only the space of the residuals $\text{PC}_{n,\text{resid}}(t)$, and we term the regimes obtained through this method *Geopotential-Jet Regimes*.

In all cases we use the principal components of each dataset and the jet speed regression coefficients computed for each dataset to perform this analysis. In this paper we use the four leading principal components, as in (Dawson et al., 2012;

Strommen et al., 2019). These explain 50% of the variance in the domain, and place focus on larger scale patterns rather than detailed regional variability, although in practice, regimes found when using a larger number of components are qualitatively identical. While regimes are identified in principal component space, either unfiltered or residual, when analysing the associated circulation patterns we always return to the full Z500 space over the Euro-Atlantic domain, by compositing the full Z500 fields using all days assigned to a given regime. Therefore composites of classical circulation regimes and geopotential-jet regimes are directly comparable.

As a note, the K-means algorithm is non-deterministic, as it requires a random-seed at initialisation which can cause convergence to different local minima. In order to assure repeatability, when we refer to 'clustering' a dataset, we run the K-means algorithm 100 times with different seeds, and use the result that maximises the inter-to-intracluster variance ratio (as defined in section 2.4). Therefore, non-determinism in the clustering method is *not* the source of the sampling variability in regime structure we identify in the following sections.

## 2.4 Regime metrics

Here we introduce the metrics used to analyse regimes in this paper. The variance ratio, regime occurrence and regime persistence are commonly found in the literature, while the regime stability and fidelity metrics are novel.

**Variance ratio** The variance ratio provides a measure of how tightly clustered regimes in a dataset are, evaluated within the space of principal components used to perform the clustering. It draws its name straightforwardly from its definition; it is equal to the variance between the regime centroids divided by the average variance between datapoints. Therefore high values correspond to reduced overlap between clusters, indicating a more clearly multimodal dataset.

**Regime stability and fidelity** In order to quantify changes in the spatial structure of regimes, we introduce two new metrics based on evaluating the average pattern correlation between regime composites computed in different datasets. Regime stability quantifies the degree to which regime patterns found within subsamples of a single dataset differ to those regime patterns found in the dataset as a whole. It therefore assesses the non-stationarity of the regime patterns, which may reflect slowly changing external drivers or sampling variability.

Figure 3 illustrates the methodology for four classical circulation regimes computed in the ERA20C dataset. First the full record is clustered, in this example 110 years, and anomaly composites of the Z500 field are produced. This procedure is repeated for subsamples of the dataset, here for three successive thirty year periods. For each, we first pair every regime pattern found in the subsample to its closest equivalent in the full dataset, using a linear sum assignment algorithm. The area-weighted pattern correlation is then computed for each of the $K$ pairs, which can be averaged to give the subsample correlation. When this subsample correlation is averaged across subsamples it gives the *regime stability*. As a correlation, values close to 1 indicate vanishing nonstationarity.

Formally, for a dataset $D$, divided into $N$ subsamples $\{d_1, ... d_n\}$, and considering $K$ regimes, the stability is given by

$$\text{Stability} = \frac{1}{KN} \sum_{k=1}^{K} \sum_{n=1}^{N} \text{Corr}(D_k, d_{k,n}) \tag{3}$$

where, we index also over the $K$ regimes. Regime fidelity is a very similar metric, but is constructed to evaluate how closely regime patterns found in one dataset resemble those found in another. In our case we always use this to compare regime patterns found within a model dataset $M$, and its subsamples $\{m_n\}$, to regime patterns found in a reanalysis dataset $R$. Again, values close to 1 indicate near perfect agreement between the model and reanalysis, while a 0 value would indicate completely unrelated regime patterns. Formally this is given by

$$\text{Fidelity} = \frac{1}{KN} \sum_{k=1}^{K} \sum_{n=1}^{N} \text{Corr}(R_k, m_{k,n}) \tag{4}$$

**Regime occurrence and persistence** While stability and fidelity help analyse spatial regime structure, the regime occurrence and persistence are used to quantify the temporal structure of regimes. Regime occurrence is defined simply as the fraction of days in a considered time-series assigned to a particular regime, while regime persistence is defined as the probability that a regime event will persist from one day to the next. Persistence is calculated by fitting a first-order Markov chain to the data. The Markovian assumption is a good fit for the regime lifetime distribution, with only very slight deviations in the fractions of events lasting $< 3$ days (see Strommen et al. (2019) and Supplementary Figure 4).

## 2.5 Defining a neutral state

Inevitably, some daily fields will not strongly resemble any of the regime patterns computed, representing transitional states or comparatively rare flow configurations. It is often useful to separate out these days, which we refer to as Neutral. This is done by calculating the pattern correlation of each day's Z500 anomaly field with regime composites computed in the full dataset, and considering only those days with a pattern correlation $\geq 0.4$ to represent active regime events.

A high threshold value is more discerning and focuses on particularly archetypal flow states, while a low threshold value increases sample size. Using data from ERA20C as a guide, we aimed to use as low a threshold as possible, while still making sure regimes were associated with changed blocking probabilities. Supplementary Figure 5 shows the frequency of spatially and temporally persistent blocking events as defined in Davini et al. (2012), for the three geopotential-jet regimes that are the focus of this work, for a range of correlation thresholds. The value of 0.4 was chosen as it is the lowest value that ensures each regime is associated with increased probability of persistent blocking at some longitude in the domain, although in the case of the Atlantic Ridge regime this is only a slight increase.

Spatial regime composites restricted to only non-Neutral days are essentially the same as those including them, as the Neutral days themselves have vanishing composites. Therefore, for simplicity of presentation we apply the neutral filtering only when considering temporal variability.

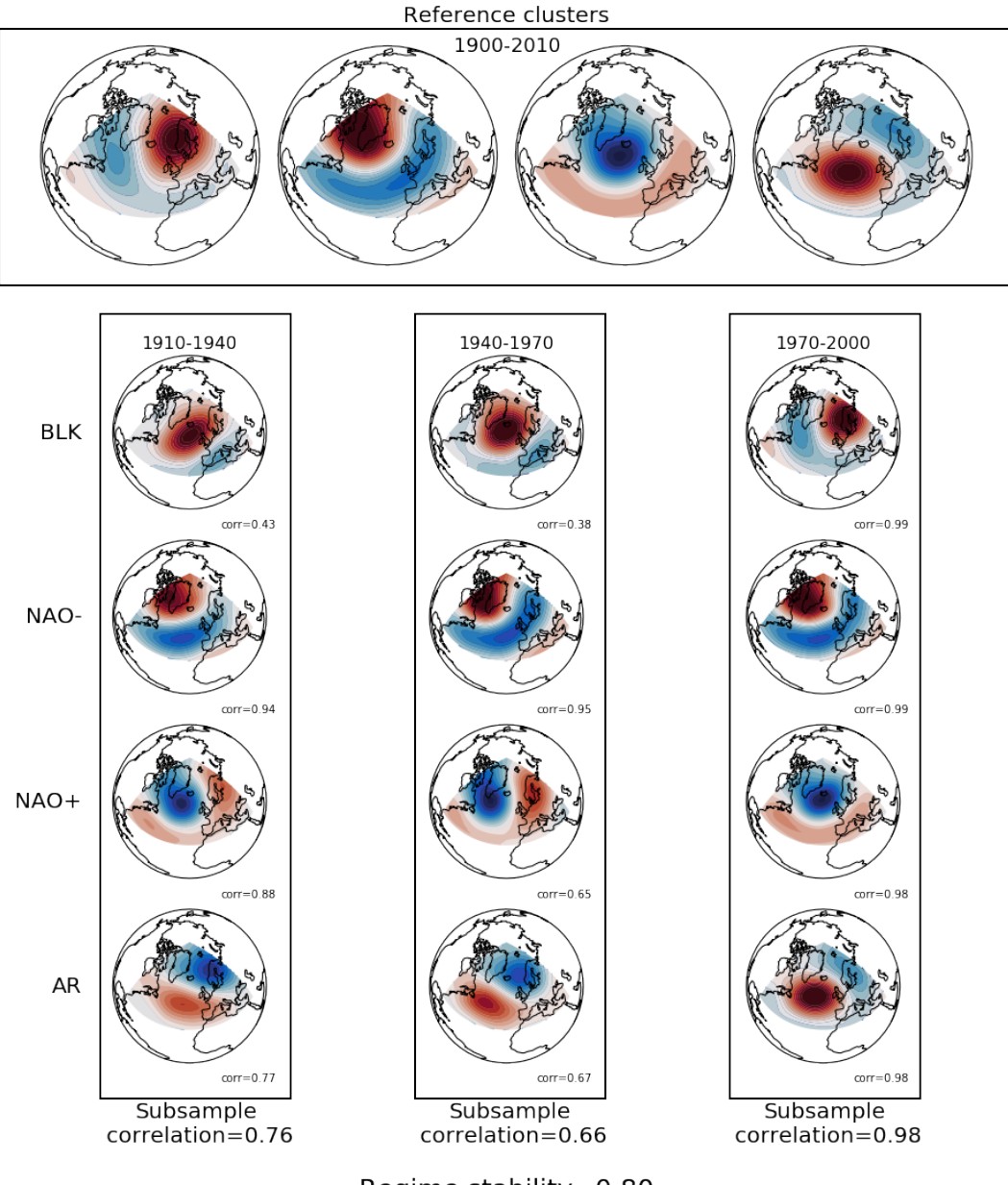

**Figure 3.** A schematic highlighting how regime stability is calculated, shown for 4 classical circulation regimes found in ERA20C. Reference regimes are obtained from the full length of the dataset, in this case 110 DJF seasons. Regimes are also calculated for a number of subsamples of the dataset covering different time periods, and the regime patterns are mapped to the reference regime with which they have the highest pattern correlation. These correlations are then averaged across regimes and subsamples to give an average measure of regime stability. Note that while only 3 independent subsamples are shown here in this illustrative schematic, overlapping time periods are used in practice to increase the number of samples.

## 2.6 Description of predictive model features

In section 5 we aim to understand the reasons that different models possess differing regime patterns. To that end, we have selected a small number of model features that have the potential to explain inter-model spread in regime behaviour. These features, which we will refer to loosely as 'model features', were chosen based on their prominence in the literature. Note that data is always interpolated onto a common 1 degree regular grid before computation (except for Gulf stream sea surface temperature gradients, see below), and that for quantities computed between 1979 and 2015, all available data in this period is used (i.e., for CMIP5 models this means 1979 to 2005).

**Jet speed** The mean daily, DJF jet speed across every available year. While the geopotential-jet regime framework views the jet speed variability as being orthogonal to regime variability, this does not exclude possible relationships between the mean jet speed and mean regime structure. For example, it is known that the mean strength of the jet is related to its latitudinal variability (Woollings et al., 2018), and this relationship may also be expected to manifest in a regime context. Abbreviation: JetSpeed.

**Arctic sea ice** The mean Arctic (40-90N) sea ice concentration in November, between 1979 and 2015. November sea ice has been extensively discussed as potentially exerting an influence on the jet and NAO on both seasonal (Deser et al. (2007); Strong and Magnusdottir (2011); Dunstone et al. (2016); Wang et al. (2017)) and climate time scales (Barnes and Screen (2015)), by acting as a source of stationary Rossby waves. In Strommen (2020) it was shown that these links are also visible in a regime context. We only consider the November means here, as these may be expected to act causally on regime variability (which is always computed using DJF data) without being immediately confounded by the fact that the winter jet also influences sea ice. While restricting to November avoids such coupling issues within a given season, we cannot rule out that the climatic mean November sea ice state is intimately coupled to the mean DJF regime variability. Two further provisos should be noted. Firstly, many studies concerned with seasonal teleconnections focus on the impact of just the Barents-Kara region: we choose here to focus on the entire Arctic, as this is common in studies focusing on longer climate time scales. A small number of models were found to be essentially ice free in November and were not included when computing correlations or ridge regressions involving sea ice metrics. They are still included for all other metrics. These are the PRIMAVERA models MPI-ESM1-2-HR, MPI-ESM1-2-XR, and the CMIP5 version of EC-Earth (all ensemble members). Abbreviation: Arctic Nov-mean.

**Eddy forcing strength** The persistence of regime events, which is closely related to the persistence of the eddy-driven jet. Many studies have demonstrated that jet anomalies are reinforced through transient eddies, either via direct forcing or eddy-meanflow feedbacks (e.g. Robinson (1996); Lorenz and Hartmann (2003)). This suggests that eddy forcing is important for regime persistence, a hypothesis which was confirmed in Strommen (2020). Inspired by the metric used in ibid, as well as the NAO-focused eddy metric of Scaife et al. (2019), we introduce in this paper a new, simple metric for measuring the strength of the eddy forcing on the jet. We compute the daily eddy momentum flux convergence of

250hPa winds:

$$E_{250} := \frac{\partial(-u''_{250}v''_{250})}{\partial y}$$

where $u''_{250}$ (resp. $v''_{250}$) is the zonal (resp. meridional), 2-6 day bandpass filtered wind field at 250hPa. Positive values of this quantity correspond to regions where the transient eddies are accelerating the westerlies (Hoskins et al., 1983). The first EOF of DJF mean $E_{250}$, resembles a northward, zonal shift of the eddies (cf. Supplementary Figure 6). Large values of the corresponding principal component $PC_{250}$ would therefore be expected to be associated with an eddy-induced northward shift of the jet. The strength of the eddy forcing is then estimated as the regression coefficient of the normalised $PC_{250}$ index against the (non-normalised) DJF mean jet latitude. This number can plausibly be interpreted as measuring how many degrees north the jet shifts in response to a unit measure of anomalous, northwards eddy activity. The $PC_{250}$ index was found to correlate strongly, but not perfectly, with the metric used in Strommen (2020). Unlike the metric in ibid, however, the regression coefficient here does not implicitly assume the existence of a multimodal jet, which is important given that not all models considered have one. Note that a consistent choice of sign for the EOF of $E_{250}$ across data sets is maintained by requiring a positive pattern correlation with the leading $E_{250}$ EOF of ERA5. Abbreviation: EF-JetLat-Regr.

**Stratospheric variability** To measure stratospheric variability, we computed the standard deviation of monthly zonal mean zonal winds at 10hPa at the equator (5S-5N). The region and pressure level were chosen so as to capture the quasi-biennial oscillation (QBO) in a simple manner. The QBO is now well resolved in many CMIP6 and PRIMAVERA models (Richter et al., 2020), unlike in CMIP5 (Butchart et al., 2018), but several models still show almost no meaningful stratospheric variability. We found that the standard deviation computed here was effective at discriminating between models: models with an unresolved QBO show almost zero monthly variability, while models with a more realistic QBO show variability comparable to that of ERA5. The potential influence of the stratosphere on Euro-Atlantic regimes has been noted in previous studies (Charlton-Perez et al., 2018; Strommen, 2020; Fabiano et al., 2021). Its importance for modulating jet stream variability more broadly is well known (Kidston et al., 2015; Domeisen et al., 2020). Abbreviation: QBO Std10.

**Gulf Stream SST gradient** The Gulf stream region exhibits a strong sea surface temperature (SST) gradient which has been suggested as contributing to regime variability by acting as a source of heat flux anomalies (O'Reilly et al., 2016). We measured this gradient by computing longitudinal means of monthly mean SSTs in the region 30S-65N, 75W-30W and computing the regression coefficient of these against latitude. Months were restricted to November through February, and latitudes are counted from south to north: the gradient thus computed is therefore always negative. The SST data is interpolated to a regular 0.5 degree grid prior to regression (with land-points masked out), in order to allow for a cleaner separation between reanalysis/high-resolution models (which have sharp gradients) and low-resolution models (which have weak gradients). Broadly speaking, this gradient may be viewed as giving a measure of the resolution of the Gulf stream, and is therefore closely linked to the strength of atmosphere-ocean coupling, with sharper gradients indicative

of potentially stronger coupling. The importance of realistic coupling for Euro-Atlantic atmospheric variability has been emphasised in several studies (cf. Small et al. (2019); Athanasiadis et al. (2020); Bellucci et al. (2021b); Zhang et al. (2021) and references therein). Abbreviation: NA SST-grad.

**Atmospheric horizontal resolution** The hypothesis that relatively high horizontal resolution may be crucial for the realistic simulation of Euro-Atlantic regimes was first raised in Dawson et al. (2012) using a single model with one ensemble member, and has since been examined in increasingly larger ensembles with multiple models using both forced SST (Strommen et al., 2019) and coupled simulations (Fabiano et al., 2020). The results of these studies are somewhat inconsistent, with each suggesting different impacts of increased resolution: the results of this paper will shed further light on this mismatch. The horizontal resolutions are measured as the approximate grid spacing at the equator (in kilometres), and were compiled by referring to tables in the CMIP5, CMIP6 and PRIMAVERA implementation papers. The resolution of reanalysis data is set, by convention, as 1km. Abbreviation: Atm Res.

To see whether these general features can predict model regime structure, we perform a multi-linear regression analysis over the whole dataset in Section 5. All model features and regime metrics are standardized to zero mean and unit variance before performing the regression. Individual missing values in the predictors (at most 5 over 70 values) have been filled with the dataset mean. For models with more than one member, the multi-member mean is considered for both the predictors and the regime metrics. Cross-correlations between the predictors are generally low (below 0.3 for most combinations), with the exception of QBO-std10 and jet speed (-0.37), QBO-std10 and atm-res (-0.32), and jet speed and EF-JetLat-regr (-0.30) (cf. Supplementary Figure 7). To assure that the partial colinearity is not influencing the regression, we adopted a ridge regression strategy, that adds a penalty for large regression coefficients (Hoerl and Kennard, 1970). A-posteriori check showed that the regression coefficients calculated in this way differ by less that 1% from those of the standard multi-linear regression.

Importantly, we originally considered a wider set of predictors, including polar vortex variability and mean North Atlantic SSTs. We used the adjusted $R^2$ metric to assess which, and how many, predictors should be used in the final regression model, and found that for a plurality of regime metrics, the 6 predictors listed above are optimal. Supplementary Figure 8 shows the impact on the adjusted $R^2$ when using only a subset of the listed predictors.

## 3 Climatological representation of regimes

### 3.1 Geopotential-jet regimes in reanalysis

In DS20, three geopotential-jet regimes, capturing a negative North Atlantic Oscillation (NAO-), Atlantic Ridge (AR) and European Blocking pattern (BLK) were identified in ERA20C as having particularly stable regime patterns over different periods of the 20th century. We review the features of those regimes here, as we find in this work that they are also stable in other reanalyses and many models, and we will be considering their representation in detail. Figure 4 shows composites of DJF Z500, U850, jet latitude and jet speed for these three geopotential-jet regimes, as computed with the ERA20C reanalysis over the period 1900-2010 (regime composites computed using other reanalyses are qualitatively identical). The AR regime

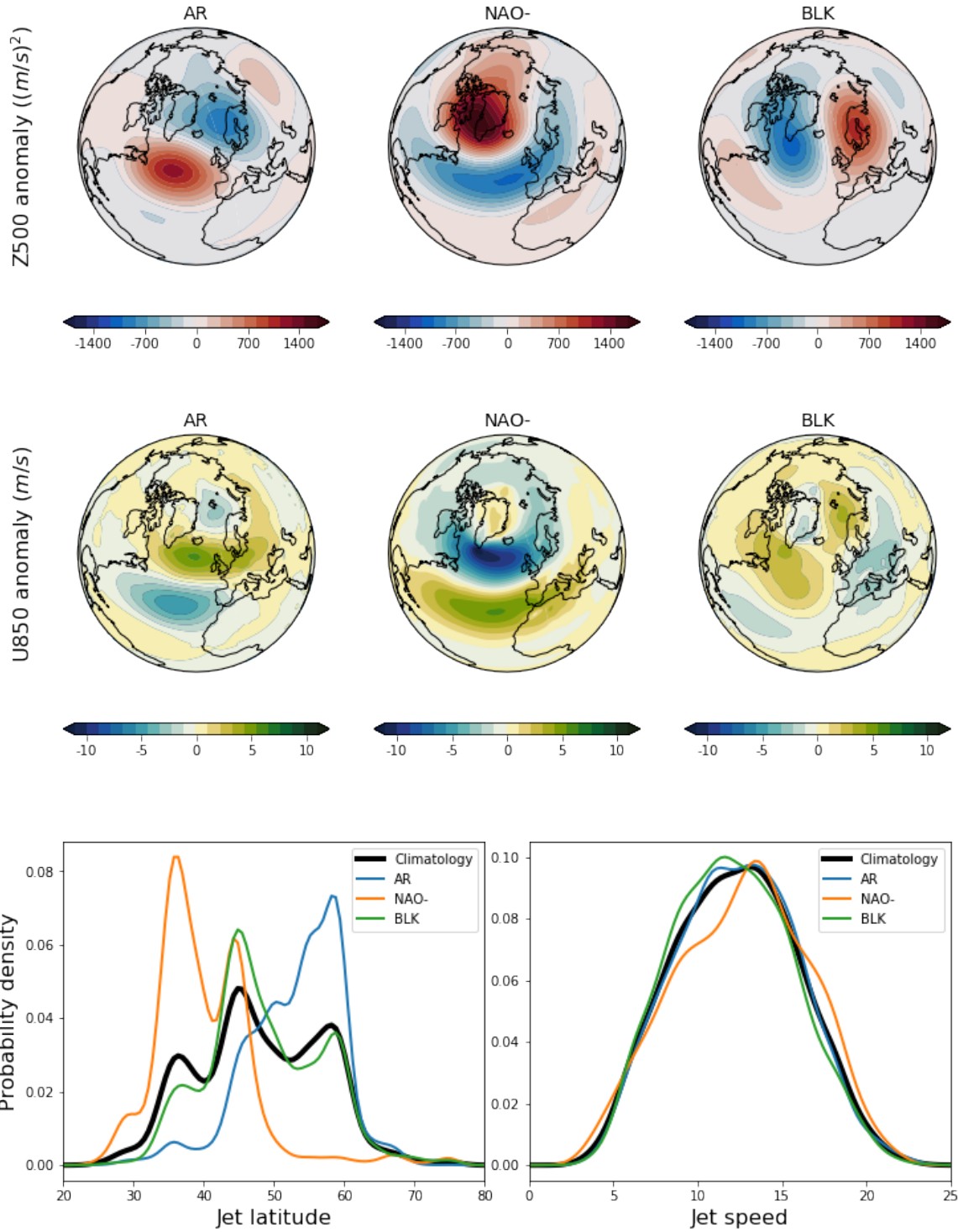

**Figure 4.** Regime composites for 3 geopotential-jet regimes in ERA20C. Top: 500 hPa geopotential height anomalies. Middle: 850hPa zonal wind speed anomalies. Bottom left: distribution of jet latitude. Bottom right: distribution of jet speed.

features a low-latitude ridge, with an anomalous low to the North-East, and tends to feature a Northerly, zonally-oriented jet latitude on the poleward flank of the ridge, with reduced wind speed on the equatorward flank. The NAO-, also referred to in the literature as a Greenland blocking pattern, features a strong anticyclone over Greenland, and low pressure in the mid-Atlantic, and a correspondingly strong reduction in wind speeds equatorward of the block. The eddy-driven jet is displaced southward, and will tend to merge with the high-level subtropical jet (Li and Wettstein, 2012). While almost all Southerly jet events are assigned to the NAO- regime, the correspondence is not one-to-one, and there are many days where a central jet is coincident with the NAO- geopotential-jet regime. The BLK pattern features a meridional dipole with a high over Scandinavia, suggesting a wavy jet structure. The BLK regime is the most weakly coupled to the jet position, featuring a trimodal latitude distribution, but it does feature a preference for central jet activity, and with the same pattern of positive poleward wind anomalies and negative equatorward anomalies as in the AR regime, the latter of which produce the low wind speeds over continental Europe which are characteristic of a European blocking.

By regressing out jet speed from the principal components prior to clustering, our regime identification prioritises features of the Z500 field that link to the spatial structure of the jet stream, rather than variations in the strength of the jet. Nevertheless, the NAO- regime does feature a small but statistically significant coincidence with faster jet speeds, indicating a weak nonlinear component to the jet–blocking relationship.

Unlike many regime frameworks used in the literature, there are no zonal flow states contained within this set of three geopotential-jet regimes. In the absence of a general definition of a regime, it is to some extent a matter of choice as to what range of flow states we wish to include in a regime framework. We take the perspective here that the purpose of a regime framework is not simply to categorise flow states but to identify dynamically meaningful recurrent states, associated with some underlying non-linearity in the flow. While historically the movement between zonal and blocked flows has often been framed as an approximately symmetric transition between two persistent regimes (Charney and DeVore (1979); Trevisan and Buzzi (1980); Legras and Ghil (1985); Molteni and Kucharski (2019)), there is far more evidence that strong non-linearities impact blocking dynamics than quasi-zonal flows. This includes the observed extended persistence of blocking (Masato et al., 2009), consistent with the posited nonlinear impacts of eddy-feedbacks (Shutts, 1983) and breaking Rossby waves (Tyrlis and Hoskins, 2008), and fundamentally nonlinear theoretical models of blocking such as Yamazaki and Itoh (2013); Nakamura and Huang (2018). Further, there is a growing understanding of the role asymmetric order-to-chaos regime transitions might play in atmospheric systems (Shen et al., 2021). For example, Faranda et al. (2016) suggests that quasi-zonal flows are best viewed as a continuum of states within an attractor basin, while Greenland blocking can be associated with intermittent excursions to the vicinity of an unstable fixed point. This is consistent with the perspective put forward in Woollings et al. (2008) that wave breaking triggers transitions to blocked flows from a default, or 'neutral' zonal state. The theoretical picture is not entirely clear however: as a counterpoint, Hochman et al. (2021) shows signs of particularly stable zonal flow regimes. Nevertheless, we conjecture, following DS20, that zonal flows are best understood through the linear variation in jet speed, while blocking regimes are best captured by clustering methods, as summarised in Figure 2. We find in the following sections that these blocking regimes offer us significant insight into model performance, justifying their use a posteriori.

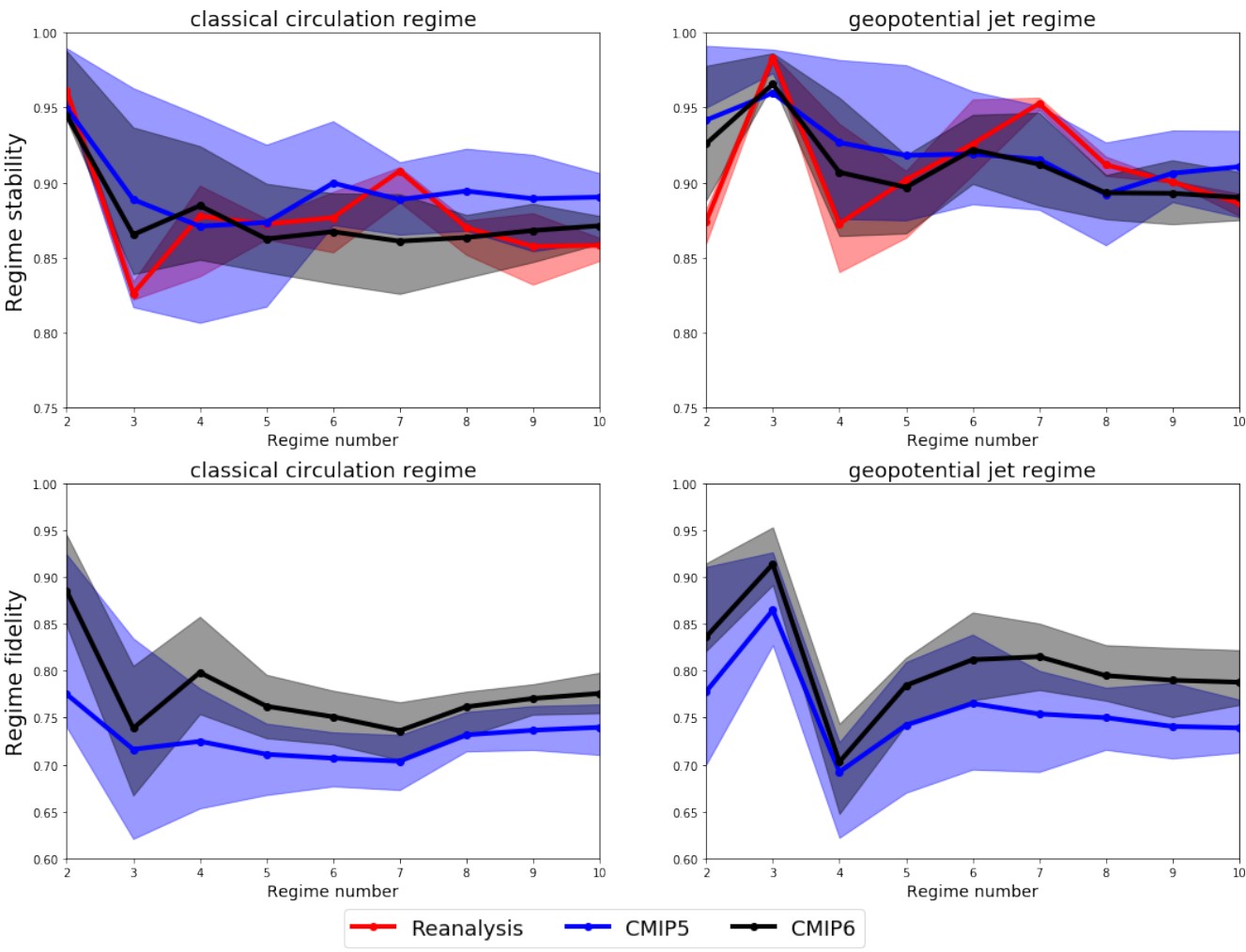

**Figure 5.** Regime stability and fidelity, calculated with 30-year subsamples, as a function of regime number, for both the classical circulation and geopotential-jet regime frameworks. Lines show the mean value for each data set, while shading shows the ensemble interquartile range.

## 3.2 Geopotential-jet regimes in CMIP5 and CMIP6

For all climate model data sets, subsamples were constructed by taking 30-year windows of DJF data at 10-year intervals, such that 4 subsamples would be computed for a 60-year dataset, and 9 subsamples for a 110-year dataset. This was done to provide a good number of subsamples for each model or reanalysis product while also maintaining a reasonable degree of independence between each sample. Contiguous windows were used rather than bootstraps in order to correctly sample any interdecadal variability in regime patterns that might be present.

The representation of spatial regime structure in reanalyses, and in the CMIP5 and CMIP6 model ensembles, is shown in Figure 5, as summarised in terms of the regime stability and fidelity metrics defined in section 2. Metrics were computed using both the classical circulation regime and geopotential-jet regime frameworks, and for cluster numbers varying from 2 to 10.

Just as was found using only ERA20C data in DS20, the regime stability in the reanalysis ensemble is maximal for 3 geopotential-jet regimes, and in most cases the geopotential-jet regimes are more stable than their classical circulation counterparts, with the only exception being the two-regime case, where the classical circulation regimes capture the two phases of the NAO with high robustness. Quite strikingly, we see that both the CMIP5 and CMIP6 ensembles show the same stability maxima for 3 geopotential-jet regimes as reanalysis, identifying them with even more confidence than the NAO dipole in the
classical circulation regime framework. That we see high stability for these three regimes, even in uninitialised climate models, indicates that this is not simply a chance feature of 20th century weather variability, but instead a property inherent to the structure of the Euro-Atlantic circulation. This validates the hypothesis and conclusions of DS20 that the regression of jet speed prior to clustering would identify more stationary regime patterns.

We note however that the extended set of 5 regimes also considered in DS20 does not appear robust when considering
multiple reanalyses and model data. Instead, a secondary stability peak is seen in reanalysis when using 7 regimes, both for classical and geopotential-jet regimes. This more detailed discretisation of the circulation produces regime patterns which are qualitatively similar to the 7 regimes of Grams et al. (2017) (patterns shown for ERA20C in Supplementary Figure 9), with only their Zonal and Atlantic Trough regimes having no direct analogue. As those regimes were shown to be highly relevant for energy applications, the set of 7 geopotential-jet regimes may be of interest for exploring interdecadal variability in surface
impacts, although this is left for future work.

There are no statistically significant differences in the regime stability of the CMIP5 and CMIP6 ensembles, but the CMIP6 models have robustly higher regime fidelity than CMIP5, showing clear progress towards a more accurately resolved Euro-Atlantic circulation. For both stability and fidelity, the spread of the CMIP6 models is reduced compared to CMIP5, indicating a greater degree of agreement, at least in this regard, between the latest generation of models. CMIP6 also follows more closely
the variations in stability seen for reanalysis, with a sharper maxima for 3 geopotential-jet regimes, and minima for 4 classical circulation regimes, than is visible in CMIP5.

The fidelity and stability of the three geopotential-jet regimes - AR, NAO- and BLK - considered separately are contrasted with their equivalents in the 4 classical circulation regime framework in Figure 6. Here we also assess the sensitivity of our results to the subsample window size used to calculate stability/fidelity, which is varied from 20 to 100 years while maintaining
the same 10-year sampling interval. The geopotential-jet regimes are robustly more stable and have higher fidelity in all datasets for all sampling windows. This is particularly pronounced in regime fidelity for the AR regime, and in both metrics for the BLK regime, while the classical circulation regime framework already captures the NAO- regime relatively well. For the geopotential-jet regimes, both CMIP ensembles have slightly lower stability than seen in reanalysis, most notable in the 20-year subsamples, and the fidelity of CMIP6 is uniformly $\sim 5\%$ higher than in CMIP5, regardless of regime or subsample
window size considered.

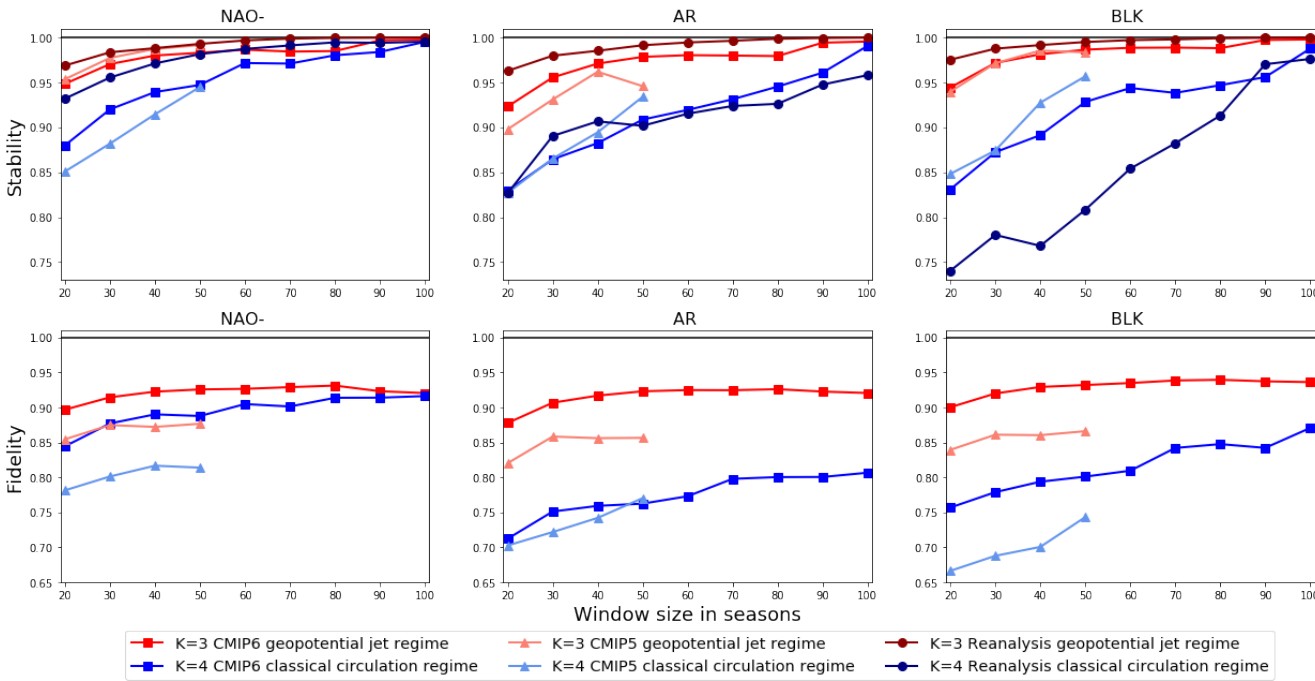

**Figure 6.** Regime stability and fidelity as a function of subsample window size. Ensemble mean values are shown for each atmospheric regime pattern as identified using both 4 classical circulation regimes and 3 geopotential-jet regimes.

## 3.3 Jet latitude regimes in CMIP5 and CMIP6

There are also substantial improvements in the representation of the jet latitude regimes themselves in CMIP6. Figure 7 shows the fraction of models in both CMIP ensembles with one, two, or three jet latitude peaks, as well as the average central latitude at which those peaks occur. While a larger fraction of CMIP6 models properly resolve the trimodal jet latitude distribution than in CMIP5, the main improvement comes from a large reduction in the number of models with unimodal jet latitude distributions. The positioning of the Southern and Central peaks is improved, and the number of peaks occurring at latitudes avoided in reanalysis is reduced, with the exception of the Northern jet latitude peak, which seems to have a degraded representation in CMIP6, shifted too far South. The Northern peak (associated with the AR regime) seems to be harder for models to capture, as most bimodal models are missing this peak (not shown). Strikingly, even in CMIP5, half the models considered produce a trimodal jet distribution, giving a less pessimistic assessment of model performance than presented in Anstey et al. (2013), which found only a few models could capture jet trimodality. This disagreement likely results from the much larger number of models included in this paper.

The full distributions of jet speed and jet latitude in CMIP5, CMIP6 and reanalysis are shown in Supplementary Figure 10. In order to holistically quantify the model error in these distributions we make use of the Wasserstein distance introduced in section 2, which measures how different reanalysis and model jet latitude distributions are, with a distance of zero indicating

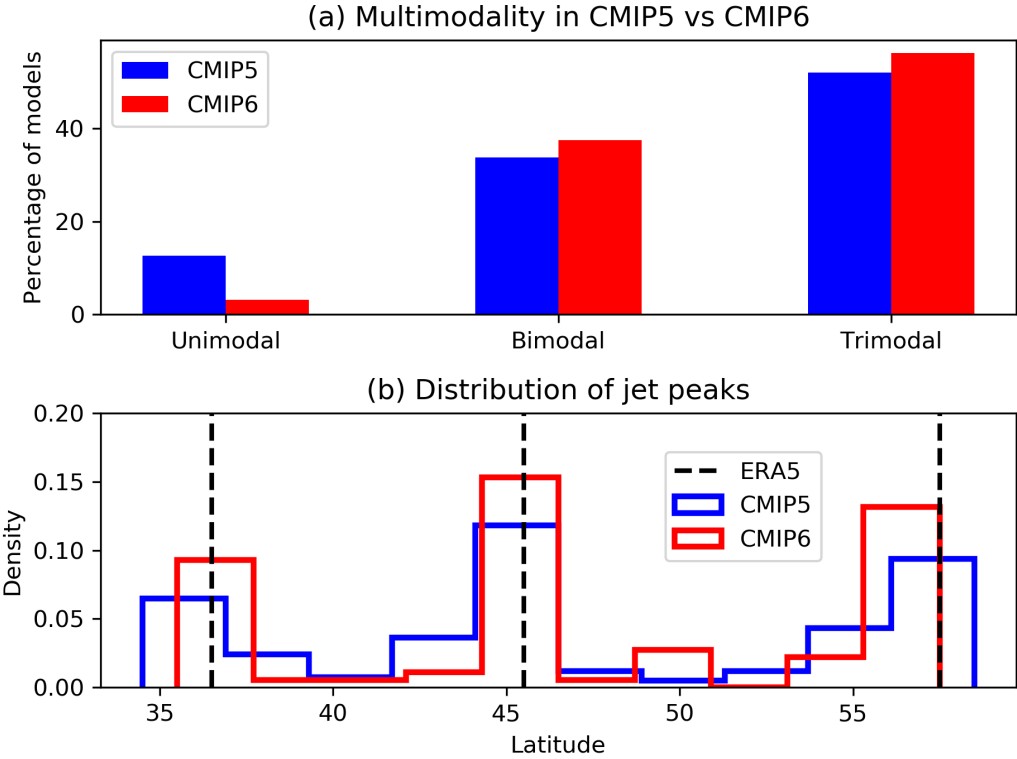

**Figure 7.** a) The percentage of climate models identified as having one, two, or three peaks in their DJF jet latitude distributions. b) Histograms of the latitude at which climate models produce jet latitude peaks, with vertical lines showing the true location of peaks found in ERA5.

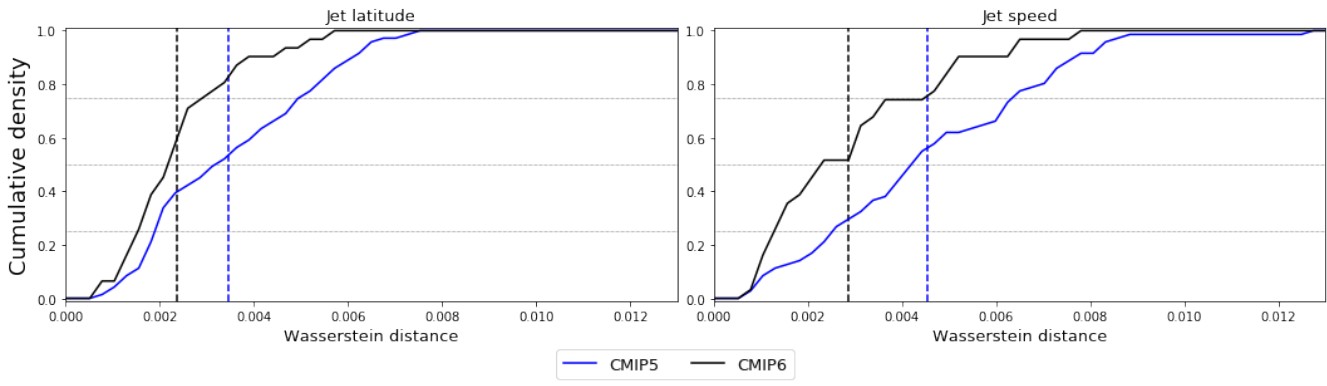

**Figure 8.** Cumulative density functions over model ensembles of the Wasserstein distance between model jet latitude and jet speed distributions and the multi-reanalysis mean distributions. Dashed vertical lines indicate the ensemble mean distances.

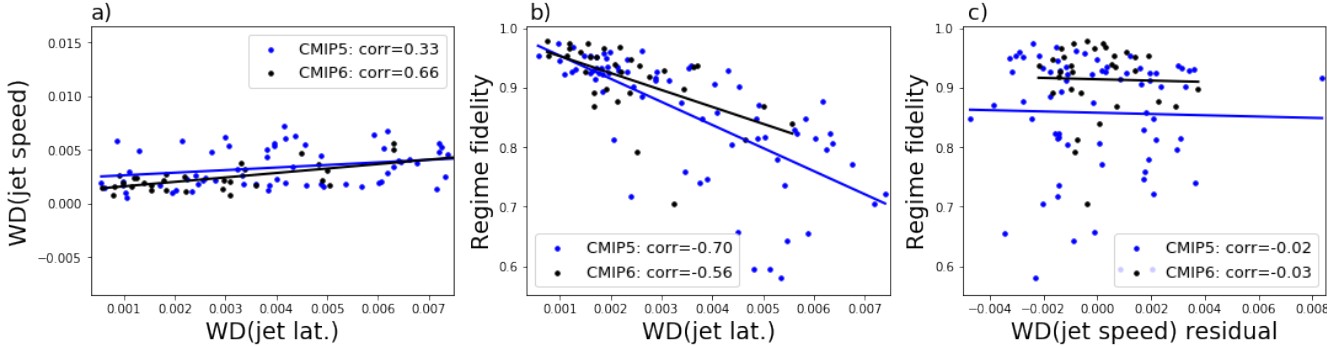

**Figure 9.** a) Correlations between the Wasserstein distances in jet latitude and jet speed in each climate model simulation. b) The same but showing correlation between Wasserstein distance in jet latitude and the fidelity of the three geopotential-jet regimes. c) The correlation between regime fidelity and the residual of the jet speed distance after covariability with the jet latitude distance has been regressed out. Correlations in panels a) and b) are significant at the 1% level, while no significant correlations are present in panel c).

perfect agreement. We can compute the Wasserstein distance between the jet latitude distribution of each model and the multi-reanalysis mean distribution, and by looking at the cumulative density of distance across the model ensemble, we can see not only if CMIP6 has lower distances (i.e. smaller errors) than CMIP5, but also if this is due to generic improvements or a reduction in outliers. We show such a cumulative density for jet latitude and also jet speed in Figure 8. The average Wasserstein

distance is 33% and 38% lower in CMIP6 than in CMIP5 for the jet latitude and jet speed distributions respectively, indicating more realistic distributions. For the jet latitude this is particularly a result of the reduction in the number of models with large errors in CMIP6, as visible in the diverging of the cumulative density functions for distances exceeding 0.002.

Of course biases in jet speed and latitude are not independent of each other, and models that do better at representing one aspect of the Euro-Atlantic circulation generally do better at representing other aspects. Figure 9a) shows a positive correla-

tion between the jet speed and jet latitude Wasserstein distances, significant at the 1% level–a relationship that is seemingly stronger in the CMIP6 ensemble. Panel b) shows that a smaller jet latitude Wasserstein distance is in turn predictive of a model having high fidelity in the 3 geopotential-jet regime patterns, with strong negative correlations between the two measures again significant at the 1% level. Jet speed errors are weakly negatively correlated to regime fidelity (corr=-0.25 – -0.4), but this can be explained entirely by the covariability with jet latitude errors; when we regress out the Wasserstein distance in jet latitude

from the Wasserstein distance in jet speed and correlate the residual with regime fidelity as in panel c), we see there is no remaining correlation. The reverse approach - computing the residual of jet latitude distance after regressing out jet speed - produces a residual which is still strongly correlated with regime fidelity (not shown). This suggests errors in jet latitude are related to errors in jet speed and in spatial regime structure, but that biases in the model jet speed do not independently relate to the quality of the anticyclonic regime structure, although we cannot rigorously assert causal relationships from a post-hoc

analysis. Results are very similar when correlating jet errors with the fidelity of the 4 classical circulation regimes, except the negative correlation between jet latitude errors and fidelity is weaker (not shown).

## 4 Historical regime variability

In the previous section we have established that CMIP6 models, and even many CMIP5 models, have an adequate representation of geopotential-jet regime patterns – the positions of the highs and lows associated with blocking anomalies. We now turn our attention to the representation of the temporal characteristics of those regimes – their lifetime and probabilities of occurrence – as well as their historical variability.

Regime occurrence and persistence, as defined in section 2 have been computed in 30-year rolling windows for all reanalyses and all CMIP models, in order to highlight the interdecadal climatic variations in the regimes. The ensemble-mean 30-year regime occurrence for each dataset is shown in Figure 10, where all models/reanalyses that have data for each full 30-year period are included in the average (Supplementary Figure 11 shows the exact amount of data included in each window). The full ensemble spread is shaded, and for the CMIP6 and reanalysis datasets, long-term mean values are also indicated.

Observational variability in regime occurrence is quite substantial, and is markedly higher for the Neutral and NAO- regime than for the AR and BLK regimes, which show less pronounced variation. The previously documented mid-century peak in NAO- occurrence that has been related to reductions in forecast skill in Weisheimer et al. (2017) is visible here, and we now also document a clear decrease in occurrence of the Neutral regime over 1970-1990. Between the reanalyses there is a fair amount of spread in the exact occurrence statistics, which is notable given the almost perfect equivalence between regime patterns found in the reanalyses. This is most pronounced in the early century Neutral and AR regimes where reanalyses disagree by as much as 3-4%, but can also be seen right into the modern era, such as in the case of the BLK regime occurrence.

The biases in CMIP6 are similar to those in CMIP5: Neutral flow states are too common and the BLK regime occurs too infrequently, both by ∼2%. There is no noticeable bias in the NAO- or AR occurrence, which represents an improvement in the NAO- relative to CMIP5. These coupled models do not reproduce the actual historical variability seen in reanalysis, and the ensemble mean occurrence is almost flat across the century, consistent with the perspective that slow SST variability – which will not be coherent across an average of coupled models – is a major driver of decadal regime variability. Analysing AMIP regime statistics could validate this claim, and is left for future work. This emphasises the importance of evaluating model regimes against an appropriately long historical benchmark. For example, looking only at the years 1980-2010, CMIP6 would seem to have a positive NAO- occurrence 'bias', while between 1955 and 1985 it would seem to have a large negative bias of 5%. Comparing centennial averages however, there is no NAO- occurrence bias at all.

Regime persistence and its variability is shown in Figure 11. Here the disagreements between reanalysis estimates of historical regime persistence are even larger than for occurrence, and worryingly divergent spread in NAO- persistence is seen between reanalyses in the first half of the 20th century. It is quite possible that this spread is a result of poor historical data constraints, which are particularly severe in the higher latitudes around Greenland. CMIP6 shows a bias towards reduced persistence for both BLK and NAO- regimes of approximately 2%, while the AR regime shows no persistence bias, and there is a tendency for CMIP6 to produce too persistent Neutral events. This is consistent with the known biases in climate models towards overly zonal flows, and underpersistent regimes, as is the lack of any substantive improvement in these biases between CMIP5 and CMIP6 (Davini and D'Andrea, 2016).

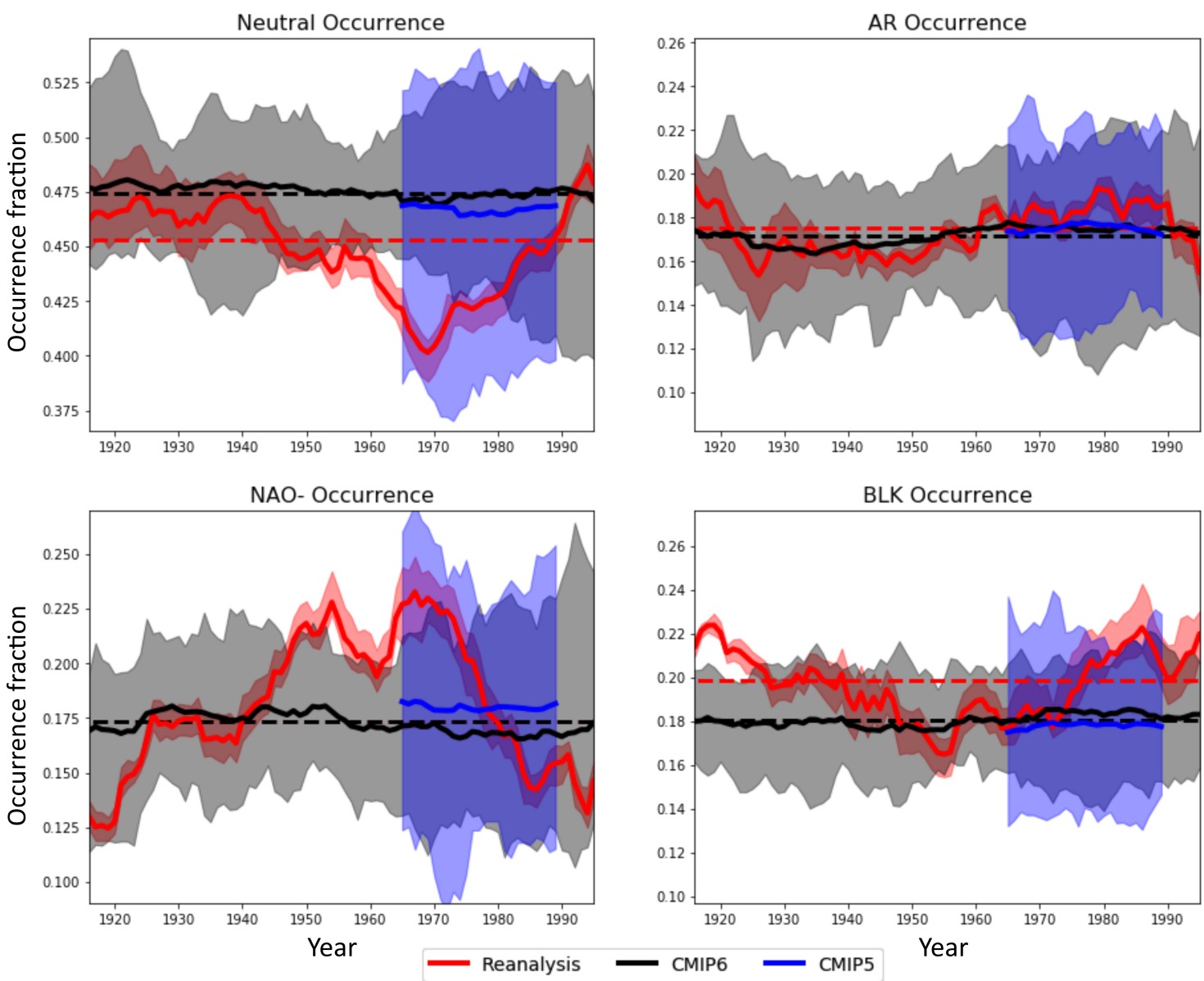

**Figure 10.** 30-year rolling windows of regime occurrence for the multireanalysis, CMIP5 and CMIP6 ensembles, with the x-axis marking the central year of the window. Thick lines show the ensemble mean, while shading indicates the spread between the maximum and minimum value in the ensemble. Dashed red and black lines show the 1900-2010 average occurrence for reanalysis and CMIP6 respectively. Supplementary Figure 11 shows the number of datasets included for each 30-year period.

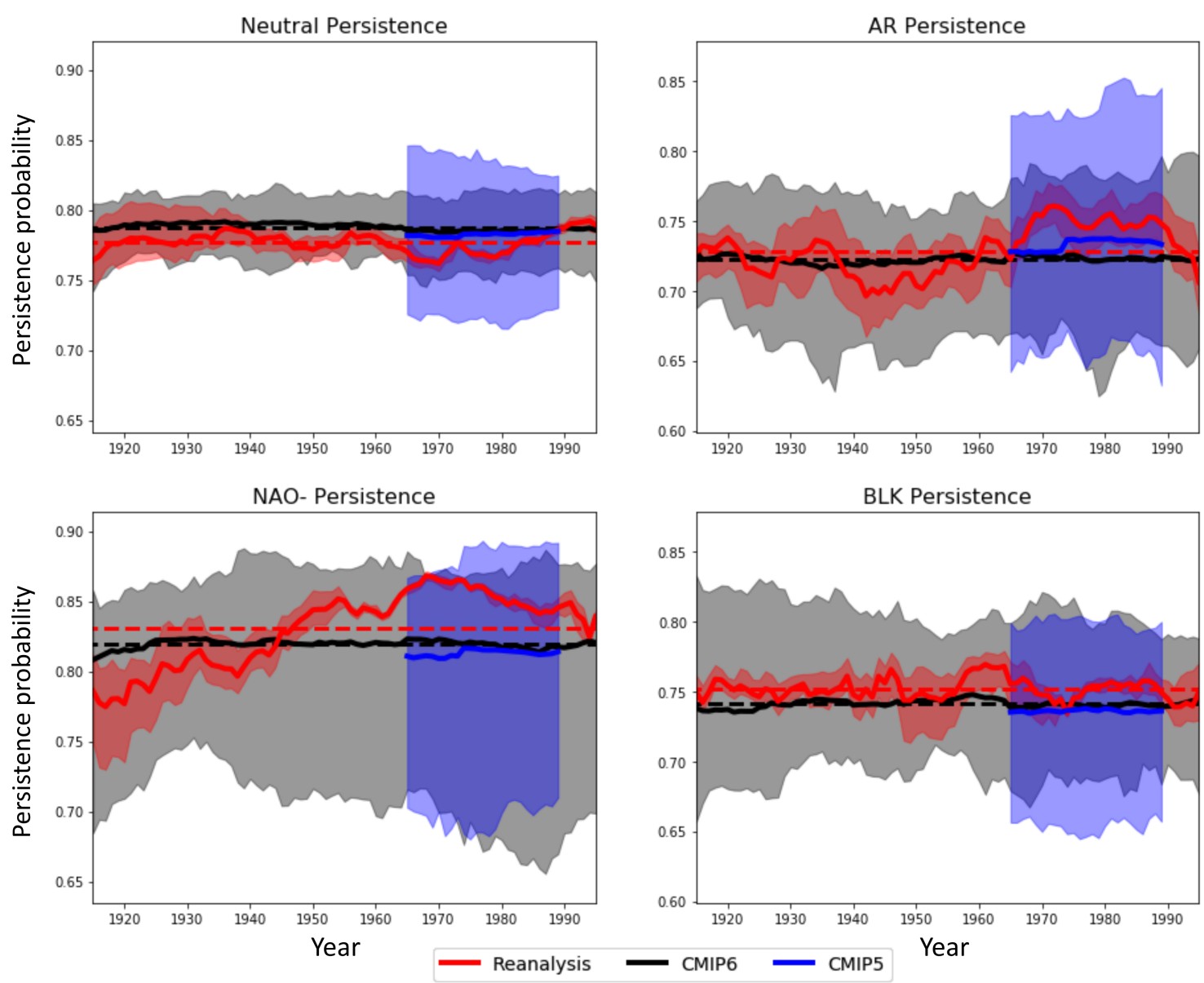

**Figure 11.** As in Figure 10 but showing regime persistence.

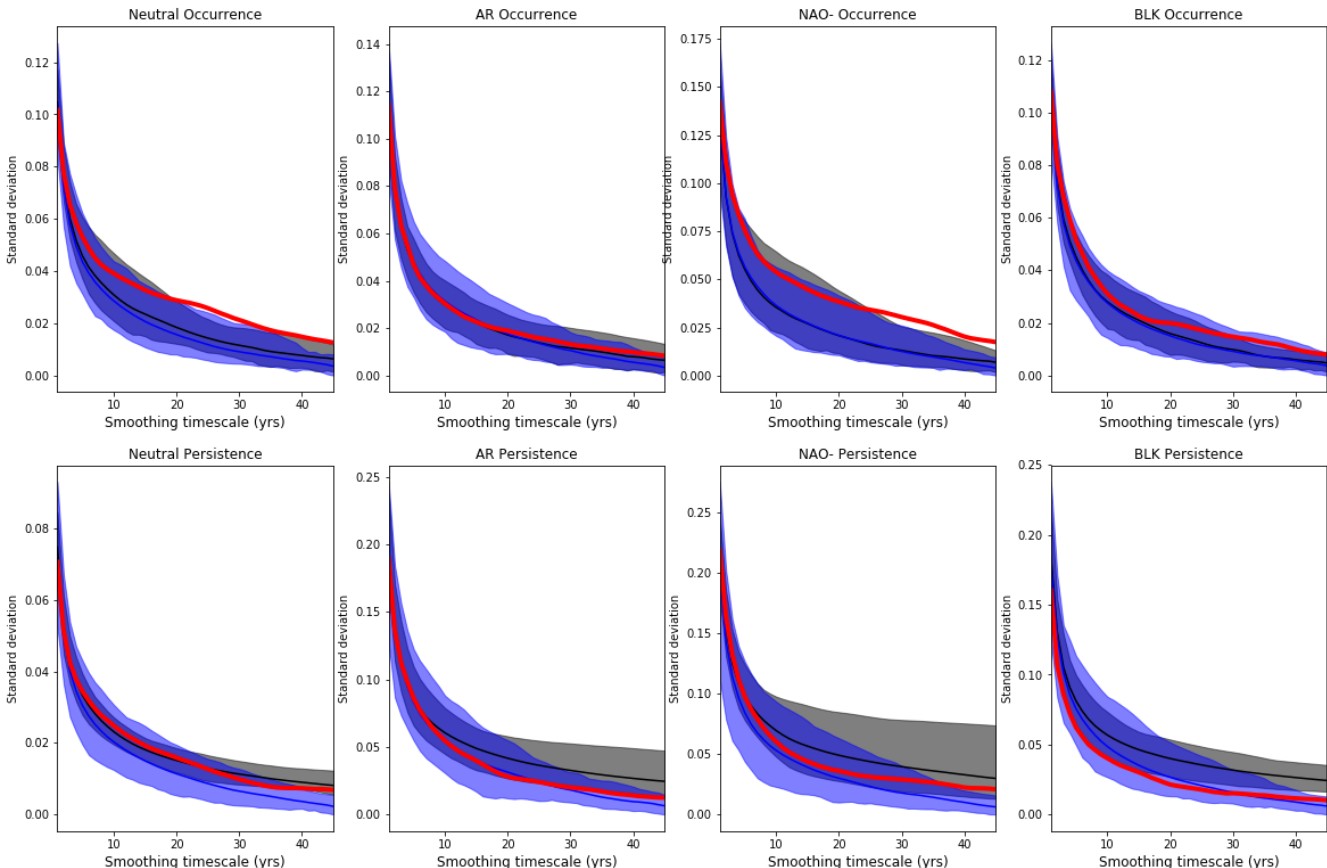

**Figure 12.** Standard deviation of seasonal regime occurrence and persistence as a function of smoothing timescale, showing the variability of the regime statistics on a range of timescales. Thick lines indicate ensemble means of each models' standard deviation, while shading shows the model spread. Red, black and blue lines indicate multi-reanalysis mean, CMIP6 and CMIP5 standard deviations respectively.

The variability of the CMIP ensemble means is of course far smaller than of the reanalysis mean, and it is also valuable to consider whether the individual models are able to represent the correct levels of slow variability in occurrence and persistence, as well as their mean values. Figure 12 shows the standard deviation in time of the occurrence and persistence metrics for a range of smoothing timescales. e.g. the 1-year smoothing timescale captures interannual variability while the 30-year smooth-
540 ing timescale isolates multidecadal variability. This approach was found to be more robust than a more-conventional Fourier spectra analysis, due to the non-stationarity in some of the reanalysis time-series, and the presence of model time series of different lengths.

The interannual variability of occurrence and persistence is well captured in models for all regimes, but biases appear on longer timescales of variability. In particular, models underestimate the inter- and multi-decadal variability of the Neutral and

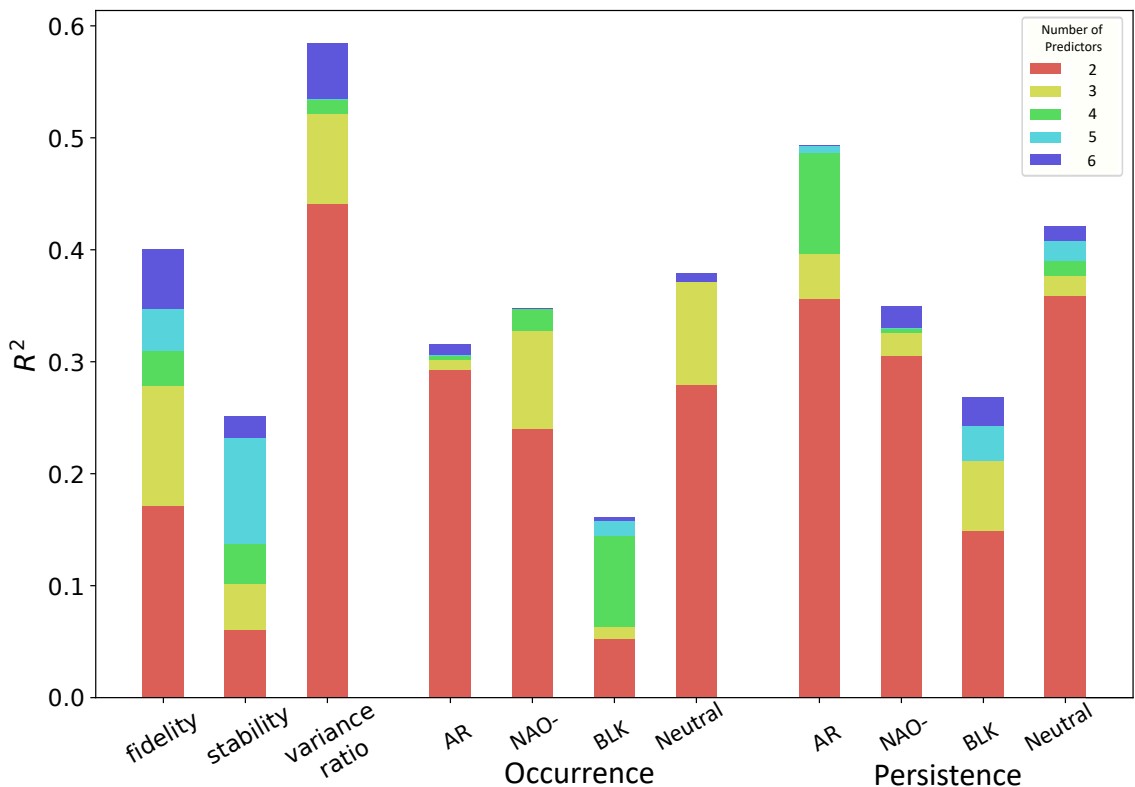

**Figure 13.** The fraction of intermodel variance in regime metrics explained by the multilinear ridge regression model, as a function of the number of predictors included.

NAO- regime occurrence, and to a lesser extent for BLK occurrence as well. Low-frequency variability in regime persistence however is too high in CMIP6, representing a notable departure from CMIP5, which has little bias in persistence variability.

## 5  Physical predictors of regime representation

In the previous sections we have explored the spatial and temporal representation of regime dynamics across the CMIP ensembles. We have seen that there is considerable variability between models in the fidelity, occurrence and persistence of regimes, and this naturally prompts us to ask *why* some models have more realistic regimes than others. In this section, we use the CMIP models as an ensemble of opportunity in order to relate regime representation to the model features specified in section 2.6, including now also the HighResMip dataset. This approach is of course imperfect: the limitations of treating CMIP datasets as true independent ensembles has been noted in Knutti et al. (2013), and there are many more differences between climate

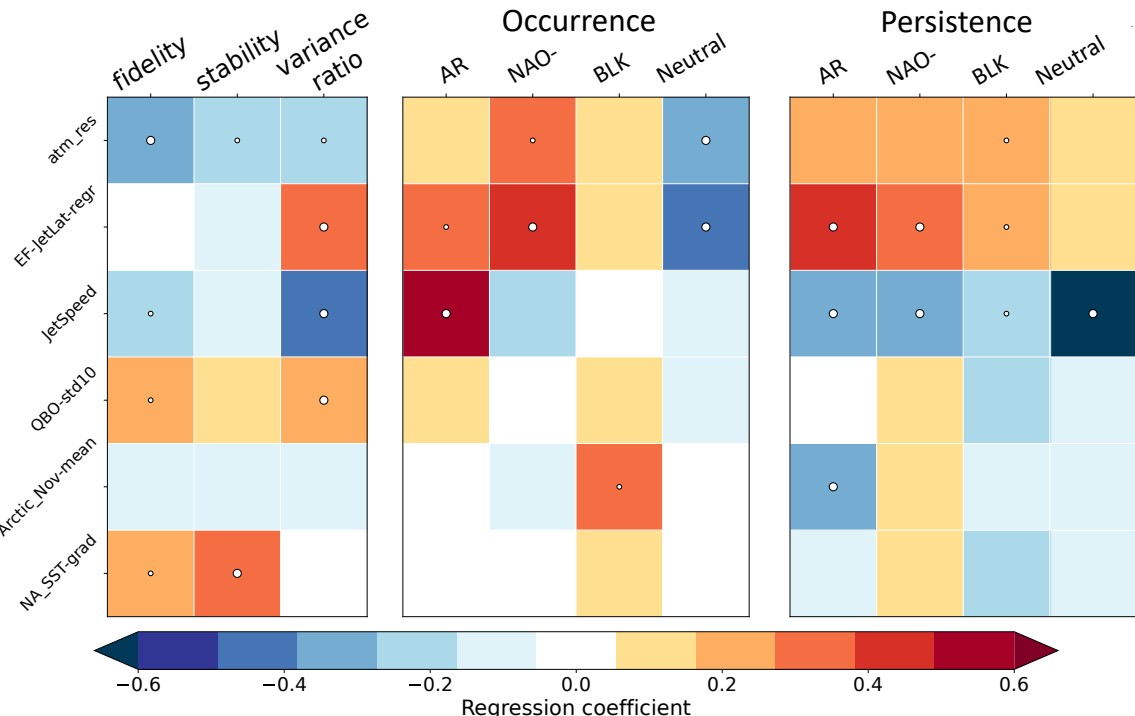

**Figure 14.** Regression coefficients from the six-predictor multilinear ridge regression model, trained on CMIP5, CMIP6 and HighResMip models, for spatial regime representation (left), regime occurrence fractions (centre) and regime persistence probabilities (right). Where the explanatory power of a predictor for a metric is significant at the 95th (99th) percentile, it is marked with a small (large) dot.

models than the small number of features we consider here. Further, a diagnostic approach cannot establish causal relation-
555 ships, which would require targeted single-model experiments with controlled parameter variations. Nevertheless, the dozens of coupled climate model integrations in the CMIP and HighResMip datasets provide tens of thousands of years of associated data, which would be computationally prohibitive to replicate in a purer fashion. The suggestive links between model features and regime representation we detect here can then provide direction for more focused modelling work in the future.

We analyse 11 metrics of regime representation: the occurrence and persistence probabilities of the 3 geopotential-jet regimes
and the Neutral state, and three spatial regime metrics: the regime stability, the regime fidelity evaluated against ERA20C, and the variance ratio. We consider both univariate correlations between model features and these regime metrics, and also fit a multivariate ridge regression model, as described in section 2.6.

## 5.1 Spatial regime structure

We start by considering the spatial regime metrics, where we have discussed the stability and fidelity in detail in sections 3 and
565 4, and large values of the variance ratio imply a stronger multimodality, and so an overall more regime-dominated dynamics.

Figure 14 shows the regression coefficients for each model feature within the ridge regression model, while Figure 13 shows the fraction of intermodel variance the ridge regression model can explain, shown also for different numbers of model features included. We see that the variance ratio is the most predictable of the spatial regime metrics, followed by regime fidelity, while regime stability is not explained so easily, with only 25% of the variance explicable in terms of the considered model features. Conjecturally, this may be because many models have stability close to 1, and so intermodel variability is lower overall (cf. Figure 6). Despite variations in predictability, all three spatial regime metrics show significant negative correlations with model resolution (remembering that smaller values of resolution are better), which can be seen most strongly for fidelity. This advances previous work showing the beneficial impact of increased resolution on regime structure (Dawson and Palmer, 2015), and on blocking as a result of improved orography (White et al., 2019; Davini et al.), now showing it holds in an essentially linear manner across the very wide range of resolutions (between 25 and 400 km) spanning HighResMip, CMIP6 and CMIP5, in a multimodel context with multiple ensemble members. Supplementary Figure 12 a) and b), which show the model distribution of fidelity and variance ratio vs resolution in detail, highlights that this span of resolutions is important for capturing this correlation robustly.

Another important feature linked to the variance ratio is the strength eddy feedbacks on the jet stream, as captured by EF-JetLat-Regr. Figure 15a) shows climate models tend to have both overly weak eddy feedbacks and variance ratios, and we see that when this bias is reduced (i.e. eddy feedbacks are stronger), models' variance ratios are closer to reanalysis. Further, both fidelity and variance ratio are significantly higher in models with lower jet speed and a more variable QBO, which, as shown in Figure 16, both represent a reduction in model bias. Low jet speeds are a natural result of increased stirring of momentum and wave activity associated with blocking (Nakamura and Huang, 2018), and so this observed correlation aligns well with theory, however the link between regime structure and the QBO is not as conceptually clear. Figure 16b) and d) show a clear group of models with very low QBO variability, primarily from CMIP5, and which upon visual inspection (not shown) were seen to be models with essentially no QBO at all. While some models without a QBO have reasonably high regime fidelity, all models with a fidelity below 0.8 lacked a QBO. This is in agreement with the well known importance of stratospheric dynamics for tropospheric regime structure (Baldwin and Dunkerton, 2001; Beerli and Grams, 2019), both due to direct forcing of the QBO on the subtropical jet, and through a forcing on the NAO modulated by the polar vortex (Gray et al., 2018).

Finally, in Figure 14 we see a weaker Gulf Stream meridional SST gradient (note that this is a negative quantity) is correlated with higher fidelity and stability in models. The link is particularly strong for regime stability, and model SST gradients are in fact the main source of predictive skill for the stability. We can understand this, somewhat conjecturally, as weak meridional SST gradients being a consequence of unresolved Gulf Stream eddies and thus a sign of weak ocean-atmosphere coupling (Bellucci et al., 2021a; Zhang et al., 2021). If the coupling is weak, then there will be less forced tropospheric variability on decadal timescales, and so less interdecadal variation in regime structure. This might be expected to result in higher regime stability. However, we are not able to assess the validity of this conjecture in this work.

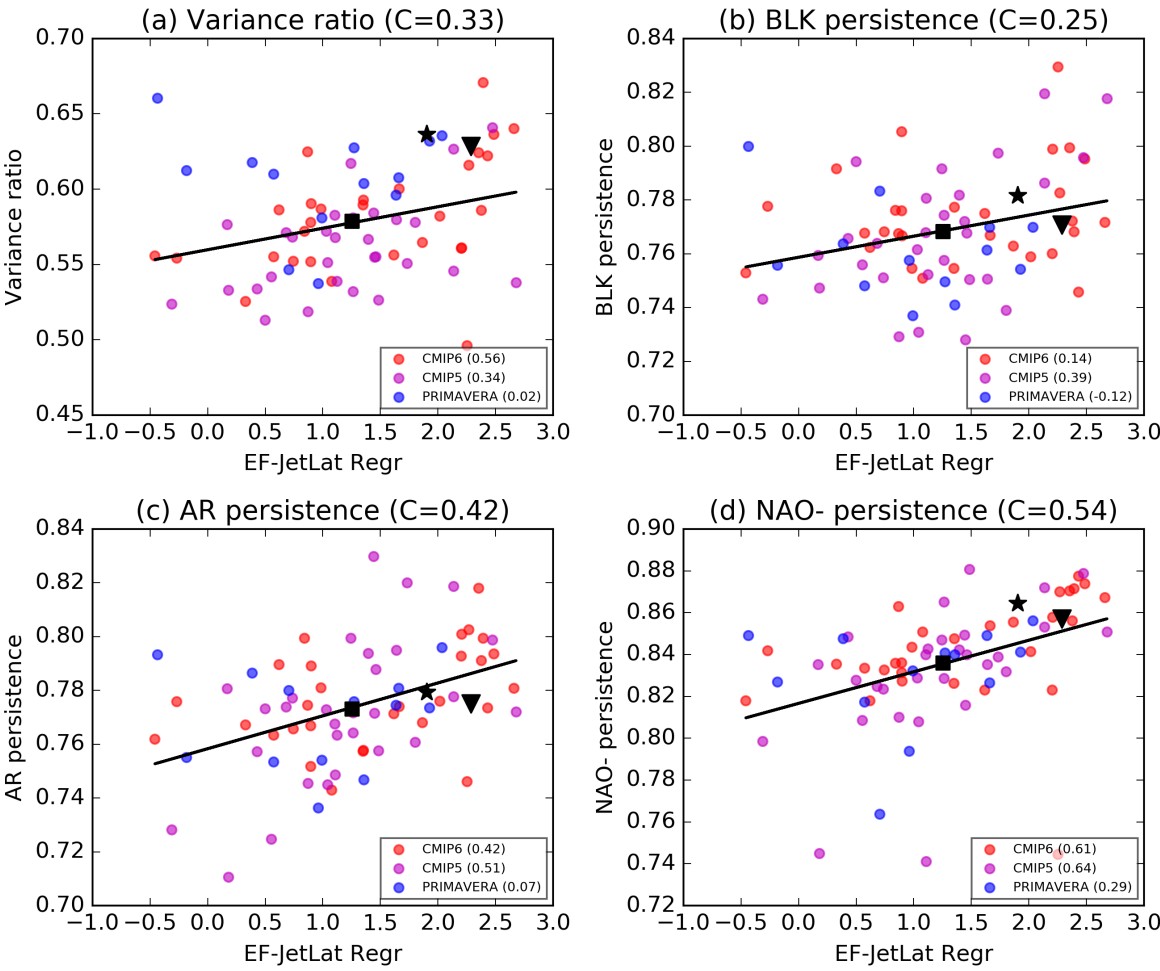

**Figure 15.** The covariation between eddy feedback strength and variance ratio (a) and regime persistence (b-d) in CMIP5, CMIP6 and HighResMip datasets. Dots mark the value for each climate model, while a square indicates the multimodel mean value. A star and triangle mark the values estimated from the ERA5 and ERA20C reanalyses respectively. For models with multiple ensemble members, the ensemble mean values have been used. Trend lines shows the best fit linear relationship for all models, with the Spearman correlation indicated as the value $C$ in each subplot title. The correlation between variables for each individual model dataset is shown in the subplot captions.

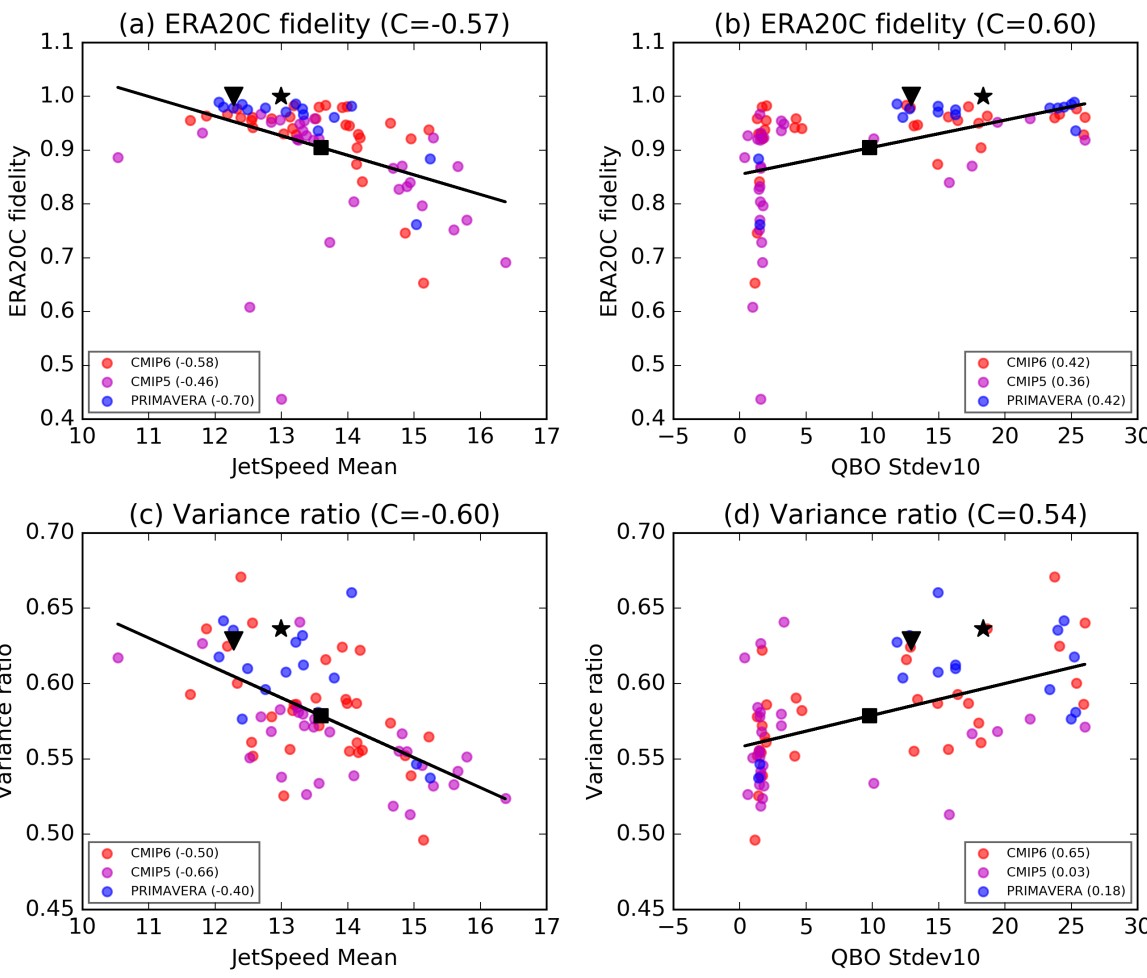

**Figure 16.** Left: As in Figure 15, but showing the covariation between mean DJF jet speed and regime fidelity and variance ratio. Right: The same, but showing relationship with QBO standard deviation measured at 10hPa.

## 5.2 Regime occurrence

Figure 13 shows that the different occurrence rates of regimes between models can also be partially explained, excluding variability in BLK occurrence. The central panel of Figure 14 shows that the strongest single relationship is the positive correlation of jet speed with AR occurrence. This was hinted at in Barnes and Polvani (2013a), which found models with faster jet speeds were biased towards a more Northerly jet latitude. Improving model resolution increases the number of Neutral days at the expense of the number of NAO- days, consistent with the elimination of the CMIP5 NAO- occurrence bias in CMIP6 (cf. Figure 10). Increasing the strength of eddy feedbacks has the opposite impact, increasing the amount of NAO- days, as well as also AR days, by reducing the number of Neutral days. Given that CMIP6 models have weak eddy feedbacks compared to reanalysis, we might expect CMIP6 to show underoccurrence biases for the NAO- and AR, and too many Neutral days. While we do indeed see a bias towards too many Neutral days, there are no corresponding biases in the AR and NAO- regimes. Instead it is BLK that occurs too infrequently in models. This may suggest the existence of a compensating model error, that we have not managed to isolate in our analysis here.

## 5.3 Regime persistence

Figure 13 shows variability in regime persistence can be more fully explained than regime occurrence, which is not obvious a priori, as persistence is generally speaking a more noisy metric than occurrence. Again, variations in BLK persistence are the least explicable. We see in the right panel of Figure 14 that low jet speed is highly correlated with high persistence across all regimes, especially in the Neutral regime, suggesting a link to longer but sparser blocking events. This is a particularly robust result, and as shown in Supplementary Figure 13 holds for each of the three model datasets considered independently. Woollings et al. (2018) showed that low jet speeds are associated with increased latitudinal jet variability, but did not distinguish between inter- and intra- jet regime variability. The increased persistence and variance ratio we see associated with low jet speeds would be conceptually most consistent with a scenario of more intra-regime variability, i.e. wobbling around a particular peak, but a more detailed investigation outside the scope of this paper would be needed to assess this. Strong eddy feedbacks are also important for the persistence of the active regimes, and the relationship is shown in detail in Figure 15 b), c), and d). This is in line with the hypothesised role of eddy vorticity fluxes as an important mechanism of blocking maintenance (Shutts, 1983). The fact that the clearest relationship between eddy forcing and persistence is seen for the NAO- regime can be understood in terms of earlier studies (Lorenz and Hartmann, 2003; Barnes and Hartmann, 2010), which show that equatorward-shifted jets are more persistent than poleward-shifted jets, owing to stronger eddy feedbacks closer to the equator. Lower levels of November Arctic sea ice also seem linked to high AR persistence. While resolution only significantly explains variability in BLK persistence within the context of the multilinear ridge regression, all regimes show a correlation between improved resolution and *decreased* blocking persistence, as seen in Supplementary Figure 12c)-f), which serves to exacerbate model bias.

## 6  Discussion and Conclusion

In this paper we have analysed how well two important aspects of the wintertime Euro-Atlantic circulation – the variability of the eddy-driven jet and transitions between anticyclonic blocking regimes – are represented in current-state-of-the-art climate models, as captured by the CMIP6 ensemble. Through comparisons with the previous generation of CMIP5 models, and including a glimpse of the next generation of models with HighResMip, we have carried out the most comprehensive analysis of model regime and jet behaviour to date, including more than 70 separate models. We have made use of the *geopotential-jet regime* framework, introduced recently in Dorrington and Strommen (2020), which filters out the linear variability of jet speed prior to clustering in order to focus on the impact of fundamentally nonlinear jet-latitude variations on the geopotential height field. With the aid of this large dataset and this hybrid methodology, we have made several key points:

- The 3 geopotential-jet regime patterns introduced in DS20 were reproduced in 5 reanalysis products and almost all models considered. They were also found to have more stable spatial regime patterns, and were better captured in models, than for any other choice of cluster number, in either of the regime frameworks considered. This makes them particularly suitable for, e.g., inter-model comparisons or analysis of slow atmospheric variability, and suggests that they are capturing a genuine multi-modality of the underlying circulation. The ability of models to well reproduce these regimes, which were adopted solely with the aim of increasing spatial regime stability, suggests that much of the previously observed poor performance of spatial regime structure in models may simply be a result of differing random internal variability in their jet speeds.

- The spatial structure of anticyclonic blocking regimes, and the distributions of daily jet latitude and speed, have significantly improved between CMIP5 and CMIP6. This is primarily due to a reduction in the number of poorly performing CMIP6 models: almost all models have a multimodal jet latitude distribution, and a majority capture all three peaks, while the 3 geopotential-jet regime patterns observed in reanalysis –associated with blocking anomalies–are more accurately observed, with average regime fidelity increasing by 6%.

- The average CMIP model has a faster jet speed and weaker eddy feedbacks on the jet latitude than found in reanalysis - related variables which indicate insufficient stirring of momentum in the jet core. Models with lower jet speeds and strong eddy-jet feedbacks were found to have stronger and more persistent regimes, indicating that these biases are probably key issues in the underpersistence of the NAO- and BLK regimes. Importantly, while models with improved horizontal resolution exhibit stronger, more stable, and more realistic regime behaviour, they tend to have slightly reduced regime persistence, and high resolution models do not have significantly reduced biases in jet speed and eddy forcing. It is clear then that resolution upgrades alone will not eliminate the biases in blocking regimes.

- Model biases in the occurrence and persistence of regimes have not improved in general, moving from CMIP5 to CMIP6. The average climate model visits the BLK regime too infrequently, and BLK and NAO- events do not persist for long enough. Neutral conditions, without a clear regime anomaly, are too prevalent. The sign of these biases is completely

consistent with the documented under-occurrence of European blocking, and underpersistent Greenland and European blocks, as identified with blocking indices in other work.

– Reanalysis shows considerable low-frequency variability in the occurrence and persistence of regimes, especially in the NAO- regime. Coupled climate models are unable to reproduce these precise patterns of variability, suggesting a chaotic origin to such variability, driven by the specific oceanic initial conditions of the real world. Just as with weather, to assess bias in the regime statistics of uninitialised models it is vital that a large range of this chaotic variability is sampled, and so it is therefore important to compare models against a suitably long historical record. Another option is to use an initialised/prescribed ocean state which should eliminate this chaotic variability and so allow biases to be identified from shorter series.

The cross-correlations between model errors in jet and regime dynamics that we have shown in this paper clearly indicate that the jet stream and blocking events are strongly interacting phenomena, and should be understood as parts of a larger coupled system of non-linear variability in the Euro-Atlantic. That is, accurately resolving blocking regimes and the trimodality of the jet are two sides of the same coin. However much historical work on the Euro-Atlantic has focused either on blocking regimes or on jet variability individually, with notable exceptions such as Madonna et al. (2017), which leaves many potentially enlightening areas of research – such as how meridional jet shifts causally relate to the onset of blocking and vice versa– under-explored.

By adopting the hybrid geopotential-jet regime framework in this paper, we have shown that stable, well-reproduced regimes can be found in climate models, which connect to both shifts in the jet latitude and to anticyclonic blocking events. The increased stability of the regimes reduces the sampling error in regime variability, and so we have been able to obtain confident estimates of model regime bias, and potential causal factors, in a statistically significant fashion. Another benefit of the geopotential-jet framework is that by considering regime variability only along the latitudinal variability of the jet, our results can be more easily related to studies considering the jet in a non-regime context. For example, the result of Woollings et al. (2018) relating the latitudinal jet variability and the mean jet speed finds a natural interpretation in our framework: no clear such interpretation seems apparent using the classical circulation regimes. More fundamentally, our framework can be viewed as a natural extension to that of Barnes and Polvani (2013b), with the latitudinal variability of the jet between peaks (the 'wobbling') interpreted as synoptic regime variability. Because the jet latitude is multimodal, we posit that the use of regime techniques may be crucial to properly account for this variability.

The improvements in regime structure we observe are consistent with improvement in classical regimes in CMIP6 (Fabiano et al., 2020). While some prior regime studies (Dawson et al., 2012; Cattiaux et al., 2013), have suggested climate models often struggle to reproduce the regime patterns found in reanalysis, we find models do fairly well at reproducing geopotential-jet regime structure. Potentially, much of the previously observed poor performance may be a combined result of the higher interdecadal variability present for classical regime patterns, and the use of relatively short reanalysis periods to compare against. When using classical circulation regimes computed with 80+ years of data, or geopotential-jet regimes computed with 30+ years of data, climate models typically achieve very high regime fidelity. The documented bias towards underoccurrence

of the BLK regime, and the underpersistence of both BLK and NAO- regimes, are consistent with biases in blocking frequency statistics (Davini and D'Andrea, 2020; Schiemann et al., 2020), demonstrating that these model errors emerge from very different methods of analysis. As long as these persistent model errors remain, there will be a corresponding degradation in the ability of models to accurately represent wintertime extremes. Concretely, with too few persistent Greenland and European blocking events in CMIP6 models, and an overestimation of non-blocked days, we might expect biases towards too few extreme

cold events (Brunner et al., 2018). and, in theory, too many extreme rainfall events (Sousa et al., 2016), although in the latter case, current generation model bias in rainfall is dominated by parameterisation errors (Prein et al., 2015).

While the impacts of resolution, eddy feedbacks, and mean jet speed on regime structure were particularly clear, we also gained some suggestive insights into stratospheric and oceanic influences on regimes. The variability of a model's QBO corre-lated significantly with stronger, more realistic, regime structure, and models without a clearly resolved QBO often - but not

always - had major deficiencies in their regimes. This suggests that regime structure is amplified by stratospheric teleconnec-tions, either directly through induced changes in the subtropical circulation, or via forcing on the polar vortex - an atmospheric feature we did not examine in this work. That some models with a poor QBO still showed good regime structure could be a result of compensating model errors, or a sign that stratospheric forcings on the troposphere can be adequately parameterised. While some significant impacts of mean November Arctic sea ice and the meridional Gulf Stream SST gradient were seen

on regime metrics, and we have conjectured possible mechanisms, the connections are still unclear and more focused work is required in the future to reveal if these are robust indeed correlations, and if so to understand the causal pathways involved. We found that our predictive model could explain more than 30% of the inter-model variance in regime representation for most metrics. The exceptions are regime stability, which was close to 1 for most models, and so did not exhibit a very large amount of inter-model variability, and the persistence and occurrence of the BLK regime. The BLK regime shows the most pronounced

model biases in CMIP6, and so understanding the origin of these biases is of key importance. Clearly, the predictive variables we considered are missing some key driver of European blocking dynamics. Given that European blocking is impacted much more by the land-processes and continental orography of Western Europe than the Greenland blocking and Atlantic ridge, it is perhaps not surprising that its behaviour is more complex to understand. One major influence on blocking we have not exam-ined here is the influence of diabatic processes, which are increasingly being recognised as important for blocking development

(Pfahl et al., 2015; Steinfeld et al., 2020), and so this may also contribute to the low explicability of BLK dynamics.

There are many avenues for future exploration, and extensions of this current paper. Having obtained a detailed understand-ing of how well climate models capture historical regimes, a natural next step is to consider future regime changes under anthropogenic warming. In future work, based upon the PhD thesis of the first listed author (Dorrington et al., 2021), we will show that geopotential-jet regimes allow for clear forced signals in both spatial and temporal regime structure to be detected.

Another important open question is whether the exact pattern of regime occurrence and persistence variability observed in reanalysis can be reproduced by AMIP models or fully initialised multidecadal forecasts. Whether such interdecadal fluc-tuations are chaotic or predictable may have important consequences for decadal forecasting and near-term climate change impacts on Western Europe. The stability metric introduced in this paper could in fact be a useful tool in future studies fo-cusing on ocean-driven regime variability, or indeed in analysing non-stationary regime frameworks. The causes of the BLK

regime bias need to be isolated before they can be improved, and the reasons for the overly weak eddy feedbacks on the jet - responsible for much regime underpersistence - need to be understood and eliminated. There is also work needed on the analysis of surface impacts associated with the geopotential-jet regimes used in this paper, and whether models capture those impacts. Finally, as mentioned, it is likely that to achieve a deeper understanding of Euro-Atlantic variability, it is important to view the eddy-driven jet stream and persistent blocking events as interconnected phenomenon, and there is still much work needed to develop a holistic understanding of how the jet and blocking covary and influence one another.

*Data availability.* The repository https://github.com/joshdorrington/GJR_hist_climate_data contains regime patterns and state sequences for all reanalysis products and models used in this paper, the predictors and regime metrics used in the regression model, and example Python code for both the computing of regime stability and the implementation of the ridge regression model.

*Author contributions.* KS and FF processed all model data, JD computed regime statistics for all datasets, KS analysed the representation of jet multimodality, and FF produced the regression model used in section 5. All authors contributed equally to the interpretation of results, and JD led the writing of the manuscript.

*Competing interests.* No competing interests.

*Acknowledgements.* – KJS was funded by a Thomas Philips and Jocelyn Keene Junior Research Fellowship at Jesus College, Oxford.

– JD was funded by NERC as part of the Environmental Research Doctoral Training Program (award number NE/L002612/1).

– FF was funded by the PRIMAVERA project of the Horizon 2020 Research Programme, under the European Commission Grant Agreement 641727.

– The analysis undertaken forms part of the IS-ENES3 project that has received funding from the European Union's Horizon 2020 research and innovation programme under grant agreement No 824084.

– We acknowledge the World Climate Research Programme, which, through its Working Group on Coupled Modelling, coordinated and promoted CMIP6 and CMIP5. We thank the climate modeling groups for producing and making available their model output, the Earth System Grid Federation (ESGF) for archiving the data and providing access, and the multiple funding agencies who support CMIP6, CMIP5 and ESGF.

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
