# Peer review of "Quantifying climate model representation of the wintertime Euro-Atlantic circulation using geopotential-jet regimes"

_Weather and Climate Dynamics, 2021_

## Author Response (AR1)

**Response to Reviewer Comments**

We thank both reviewers for their useful and detailed comments. Both reviewers expressed some doubt over some of our methodological choices and some of our interpretation of results. Specifically, the importance of stationary regimes was questioned, and the logic of why geopotential-jet regimes were particularly useful for model analysis was unclear. As these are fundamental premises on which our work is based, we believe this represents a failure on our part to sufficiently discuss the problems with unstable spatial regime patterns, and the current status quo in the literature on circulation regimes. Therefore we have completely rewritten the introduction to provide a clearer and more pedagogical discussion, which we believe will now address these questions of interpretation and motivation.

The reviewers also pointed out that the methods and section 5 were not as clear as they could have been, and these have been substantially redrafted, and rewritten respectively, for increased clarity. The results themselves remain unchanged, although we have added one additional schematic figure (the new figure 1), introduced additional SI figures 1, 2, 9 and 11, and moved the previous figures 16 and 17 into the SI as recommended by reviewer 1. We now address the specific comments of each reviewer below:

**Reviewer 1**

There is a broad literature on the nature of blocked versus zonal flows in the North Atlantic, and more broadly the midlatitudes, from the seminal works of Charney and DeVore in the late 70s to more recent theoretical perspectives (e.g. Faranda et al., 2016). Here the authors define three blocked regimes, and overlook entirely the rich discussion around the existence or lack thereof of zonal versus blocked equilibria in the mid-latitude flow, and the work on zonal flow regimes. While I still believe that the analysis of the regimes defined by the authors may be useful, it is less general than the "Euro-Atlantic Weather Regimes" they claim in the title, and may more aptly be called an analysis of blocked flows in the North Atlantic.

*We have now extended section 3.1 to include a discussion of the absence of zonal regime patterns in the geopotential-jet regime framework, and our justifications for this, beginning on Line 409. The title has been altered to 'Quantifying climate model representation of the wintertime Euro-Atlantic circulation using geopotential-jet regimes' which avoids the use of the term 'Weather Regimes', and which, upon reflection, we feel is more descriptive in any case.*

Related to the above, a general criticism that could be moved to the rationale of this study is the idea put forth by the authors that time invariance of regimes is always a desireable feature of regime definition. This then leads the authors to only identify blocked regimes. One could well imagine that information provided by the change in the regimes in the

historical period may contain useful dynamical information and be used to evaluate model behaviour (perhaps with the support of AMIP simulation), while the future changes in regimes may be useful to compare different models between themselves. In this study, the authors wish to obtain time-invariant regimes, but that comes with the hidden assumption that the system the regimes are computed on is sufficiently stationary over the analysed timeperiod for regime stability to be a positive and not a detrimental feature of the regimes.

*We consider addressing this point to be of central importance to our work, and have considerably restructured our introduction to provide a clearer exposition of our motivation and approach. Importantly, we feel invariance is needed for many practical applications, and is commonly assumed by most regime analyses. In fact, by introducing the stability metric we actively assess the possibility of non-stationarity. Even though we focus here on regimes that have stable spatial regime structure, for reasons we hope are now clearly explained, the same stability metric could be very useful for studying non-stationary regimes as well, although as we now discuss there are non-trivial technical issues to solve to perform such a study well. We comment on this point also in the conclusion on line 730.*

451-460 I find this section a bit chaotic. First you introduce the figures in block, and then you discuss some of these in detail (but not others) in the various sub-sections. As the paper is quite long, if this part is not essential I would suggest that the authors remove it and introduce the figures when they are being described in each subsection, perhaps moving the figures that are not commented on in detail to the Supplement. Also, throughout Sect. 5, the authors could state explicitly "(not shown)" when in the text they comment on some aspect of their regression model that is not explicitly illustrated in the figures. There are also several statements in Sect. 5 that are supported by the figures but that are not followed by a reference to any figure.

*This section has been completely restructured to provide a more natural flow, and to link each statement made to a particular figure. Figures 16 and 17 from the original manuscript have been moved to the SI*

18 Did the authors actually intend to highlight the North Pacific here?

*The reference to the North Pacific was indeed intentional, simply to address a common question from non-specialists as to why the North Atlantic has pronounced regime behaviour but the North Pacific doesn't. However the point was parenthetical to the main thrust of the paper, and has been cut as part of the restructuring of the introduction.*

68-69 I am not sure I follow the logic whereby these regimes being well-represented in climate models make them well-suited for model evaluation. The logic risks being somewhat circular.

*The key point here is that if models have similar spatial regime structure (i.e. similar anomaly patterns associated with each regime) then it is meaningful to compare their temporal statistics, and equally, if they have spatial structure close to reanalysis regimes then we can meaningfully discuss their biases in temporal regime structure. This is now addressed in the rewritten introduction in detail, specifically in the paragraph beginning on L145.*

86 How do you obtain 1 degree data from the reanalyses that have a lower native resolution?

*1 degree data was obtained by linear interpolation, as is now mentioned on L170. Early versions of this work investigated using 2.5 degree data were found to be qualitatively identical.*

93 It is odd to refer to table S3 before table S2. Perhaps swap the order of these?

*The order of these tables has been switched.*

119 This is not true, and one could easily devise a distance metric where this does not hold.

*This sentence (now on L210) has been rephrased to be more precise.*

130 Is 4 EOFs not a very restrictive number? What fraction of the variance do they explain? Many studies, including some the authors cite, use more EOFs as basis for the clustering. I appreciate that this classifications has been previously used by the authors, but some further details on how it reflects intraseasonal variability should be provided.

*4 EOFs explain ~50% of the variance. In practice the results are nearly identical when using a larger number of EOFs, which we now mention in text (starting L234). For example, DS20 uses 10 EOFs and obtains regime patterns and stability results for ERA20C which are difficult to distinguish from those presented in this work.*

144-145 Could you explain better what role the maximisation plays here? One could imagine simply computing an area-weighted average pattern correlation between the matching regimes in the two datasets as a comparison metric. Or do you mean that you identify the matching regimes as the pairs with the maximum pattern correlation? In the latter case, I would recommend rephrasing the sentence to state this more clearly.

*The latter point is indeed what is meant here. The sentence has been rephrased to increase clarity (now L260)*

176 This is perhaps a detail, but is the blocking in the AR regime for a threshold of 0.4 significantly above average? The difference shown in Fig. S3 relative to climatology looks to be minimal.

*You're right that there is very little deviation for the AR regime in figure 3 for a 0.4 threshold. The neutral state threshold is now discussed on L287-296, and a more accurate description of Fig S3 is now included.*

228-229 It is odd to have this bit about the sign of the EOFs here. Should you not mention this somewhere in Sect. 2.3, perhaps when you introduce pattern correlation?

*This sentence refers to the EOFs of the eddy momentum field, not the EOFs of Z500 used to define regimes. This has been clarified in the text (L333).*

259 Do the authors intend 1 degree here?

*No, 1km was intended. Ultimately assigning a resolution to a reanalysis product is slightly arbitrary. As reanalysis values are being plotted in figures S12 and S16 purely for visual reference, we made a choice of 1 km, so that the reanalysis values could be clearly seen. This is why we say 'by convention, 1km'.*

Starting l. 265 Could you not state this in simpler terms, i.e. that using adjusted $R^2$ and Ridge Regression you find that 6 predictors is the optimal number, and that the 6 predictors you list above are in turn the optimal ones? Also, does the optimality of the chosen predictors hold for all regime behaviours considered in Fig. S6 or only for a subset?

*This section has been rewritten to increase overall clarity. Indeed, 6 predictors is not optimal for every single regime metric, but only a plurality of them. This is now explicitly mentioned starting on L383.*

Methods: You should specify how you identify blocking. The description in the caption of Fig. S3 does not allow for reproducibility.

*Blocking was identified using the 2D blocking event index of Davini 2012. This is now stated on 289 and in the updated SI figure caption (S5).*

315-319 Here you could note that Grams and colleagues have indeed recently adopted a 7-regime definiton, and have also argued that fewer regimes may not sufficiently describe the intraseasonal weather variability in the region.

*This is a good point, and we now mention the connection to the work of Grams 2017 on L448, and engage with the point concerning intreaseasonal variability. An SI figure of the relevant flow patterns is also now included for reference (S9).*

322-323 Is it really the case? If you plot the difference between CMIP5 and CMIP6 in the lower RHS panel of fig. 4, does 3 actually stand out?

*Thank you for calling this point to attention. Upon closer examination, this does not fully hold, as seen in the plot below. As the importance of this point was ultimately quite minor, it has been removed.*

[Figure]

324-326 Does this claim not contrast with the fact that CMIP6 models have higher fidelity for essentially all regime numbers?

*Again, this relatively minor point has now been cut, but there is no contradiction: arbitrary partitions of two similar, continuous distributions are more likely to be similar than arbitrary partitions of two different distributions. As the general representation of the Euro-Atlantic improves, so should the 'base' fidelity of an arbitrary clustering. However if regimes really exist in the system, you would expect those to improve much more, and there are some signs of this in the above plot.*

346 But by this same metric the location of the Northern peak seems to have become worse?

*This is correct. We now comment on this explicitly, on L474.*

408 I am not sure I understand how Fig. 11 shows something about "whether the individual models are able to represent the correct levels". The authors still show bulk statistics, much like in Figs. 9 and 10.

*Figure 12 (11 in the original draft) shows bulk statistics, but of the standard deviation of regime occurrence and persistence, and so does indeed capture the levels of multiannual variability present in each dataset.*

412 Analysus --> analysis

***Corrected.***

447 Please introduce the variance ratio in the methods, especially as you also mention this in the context of the "classical" k-means clustering regimes, which may cause confusion for some readers.

*This is now included on L249.*

507 "persistence (the jet wobbling more within distinct regimes)" I am not sure I follow this logic. A priori, one could expect some of the regimes to be more persistent when characterised by a low latitudinal jet variance. If all regimes, when persistent, are associated with large latitudinal jet variability, then clearly the identified regimes reflect only a specific aspect of "the longitudinal and latitudinal variability of the jet" that the authors mention in the introduction.

*Thank you for raising this point. Upon closer reflection we believe we were overinterpreting our data in this paragraph. We were thinking that increased regime persistence would imply that variability was more internal to the regimes, rather than representing inter-regime transitions. However, there are indeed less intuitive but equally plausible alternatives. The original comment has been cut and we make a more qualified point on L618.*

27 The last point in the list, which supports the choice of analysis framework, should perhaps come first (or at least closer to the top of the list)?

*This has been moved to the top of the list*

568-570 "The latitudinal variability of the jet (the 'wobbling') interpreted as regime variability." Does this not contradict what you stated in Sect. 5.3 ("our result therefore adds additional clarity by demonstrating that the extra variability primarily projects onto regime persistence")?

*Here there is a distinction to be made between synoptic timescale variability (the context in which we refer to wobbling of the jet) and multidecadal or intermodel variability (the context for the statement in section 5). This has been clarified, now on L688*

583 "too many extreme rainfall events". Usually the models have the opposite problem, namely that they do not produce enough extreme rainfall compared to observations.

*As rainfall is parameterised in all current climate models, parameterisation errors in cloud physics, as well as thermodynamic components, will dominate rainfall bias. Talking in terms of regimes, we are here only looking at the large scale dynamical setting, hence the inconsistency. However it is possible the dynamical component will become more important as the community moves towards convection permitting models. We have qualified this sentence accordingly, now on L704*

Fig. S3: There is an "?" in the caption.

*The missing reference has been included.*

Fig. S3: "For a threshold of 0.4 the neutral regime features less blocking than in the overall climatology at all longitudes". Is this really the case? It is sort of hard to tell from the figure, but it does look as though this does not hold around 10E?

*Thank you for pointing this out. The caption now reads "or a threshold of 0.4 the neutral regime features less blocking than in the overall climatology at the majority of longitudes, excluding [10E-5W], while all active regimes show more blocking than in climatology, within particular longitude ranges."*

Figs. 9 and 10 Please label the axes.

*These figures (now 10 and 11) are now labelled.*

Figs. 9 and 10: The number of models included in the different years will affect the spread shown in the plots. Can the authors indicate, either in the figures themselves or in an SI figure/table how many models were included for every year?

*Table S4 now shows the number of models available for each year, while figure S6 shows the number of models included in each 30-year average, shown in what are now figures 10 and 11. This is mentioned in the caption of figure 10.*

Fig. 11 Caption: "mulit" --> "multi"

*Corrected.*

Fig. S8 and following: please check the captions to see whether they refer to the correct figures in the main text.

*These have now been corrected*

**Reviewer 2**

I wonder whether the title can be improved. I see that the study focuses on the phenomenology of regimes and not on their dynamics.

*We have changed our title to the more descriptive 'Quantifying climate model representation of the wintertime Euro-Atlantic circulation using geopotential-jet regimes'*

In the abstract you refer to 'wintertime circulation' and then mention that this is dominated in the Atlantic by non linear flows. Please be more specific here, for instance mention you look at low frequency atmospheric variability.

*We now mention low frequency variability specifically in the abstract.*

Line 15, complex orography and external forcing are not a feature of the wintertime atmospheric circulation, please refer to an interaction with them.

*Corrected*

Table 1 please clarify what is meant with 'experimental'.

*Here what is meant is that the GFS cycle used to produce the reanalysis does not correspond to any operational forecast cycle. A note has been added to the table to make this point.*

Line 84 why use one ensemble member? Is the ensemble member generated with a lower resolution compared to the reanalysis ( I think for ERA5 this is the case)?

*We use the actual ERA5 analysis, not one of the 10 EDA members. We were referring to reanalyses with no privileged member, such as CERA20C. We now clarify by saying: 'In the case of reanalyses with multiple equivalent ensemble members, we always use the first member' (L168)*

Lines 111-119 this is unclear, please provide more details about the methodology and its interpretation.

*The methods has been substantially rewritten to add clarity, the introduction of the Wasserstein distance now includes a precise formulation which should clarify the interpretation (L 204)*

Line 130 I was surprised by the use of 4 EOFs, form literature I would expect a higher number of EOFs. Is this common to other studies? Do the authors have an idea of how this choice can influence their results?

*4 EOFs explain ~50% of the variance. In practice the results are nearly identical when using a larger number of EOFs, which we now mention in text (starting L234). For example, DS20 uses 10 EOFs and obtains regime patterns and stability results for ERA20C which are difficult to distinguish from those presented in this work.*

Line 136-143 Again please provide more details on the methodology.

*This is now discussed in more detail , starting on L225*

Line 160 consider giving an explicit definition of stability and fidelity.

*Explicit formulas for both are now included in the methods starting on L265*

You introduce a set of predictors that mix Eddy forcing and atmospheric horizontal resolution with remote drivers. I disagree that eddies and resolution can be labeled as predictors and put on the same level of other drivers.

*We now refer to these 6 quantities as model features, and qualify our use of the term 'predictors'. Here we are only trying to claim that these general model features can be used to predict the regime behaviour, rather than assert explicit causal links.*

Line 314 please report in the text the conclusion of DS

*This is now included on L445.*

You should mention that the 7 regime approach has been used elsewhere in scientific literature.

*The important work of Grams 2017 is now discussed, on L449*

Line 329 the 'general convergence in ensemble behaviour' is rather vague, please explain.

*Clarified on L456.*

Line 345 but the northern regime latitude is actually worse in CMIP6 (?)

*This is correct. We now comment on this explicitly, on L474.*

The construction and discussion of figure 7 is a bit hard to follow.

*The discussion of what is now figure 8, on L482, has been clarified . Hopefully in combination with the improved introduction of the Wasserstein distance in section 2, this should now be clear.*

Caption of figure 9, what is the 'full ensemble spread'?

*Here we plot the highest and lowest value present in the ensemble for each time window. This is now made explicit in the caption of figure 10.*

I understand that the study is based on the assumption that historical simulations should reproduce the observed behaviour of fixed regimes in the reanalysis. While this is certainly a reasonable exercise, the hypotheses introduced by the authors should be clearly discussed.

*This is now discussed extensively in the rewritten introduction.*

Line 543 this point is rather obscure, consider rewriting it.

*This paragraph starting on L665 has been rewritten to increase clarity.*

Line 550 I am not sure I can buy the point of the suitability for intermodal comparison. Please put forward the argument with a broader discussion or consider dropping it.

*We consider this an important point, and have motivated our reasoning in the restructured introduction. We have also expanded the discussion on this on L644.*

---

## Author Response (AR3)

**Response to minor revisions**

We'd like to thank both reviewers again for their time and considered comments on both drafts of this paper, we feel they have helped significantly strengthen our presentation of the work. Here we respond to the individual minor revisions requested in the second round of review. A small number of typos have also been corrected.

**Reviewer 1**

1. I appreciate the discussion the authors have added concerning the issue of zonal vs blocked flows in Sect. 3.1. As a further suggestion, I would recommend streamlining the terminology they use. Right now, they adopt both the zonal/blocked and cyclonic/anticyclonic terminology in the same paragraph, which may confuse some readers not familiar with this specific topic.

*The language in section 3.1 has now been streamlined to use the language of blocked and zonal flow states almost exclusively.*

2. I have missed where the authors discuss how their stability metric could be used to study nonstationary regimes. Please ensure that this point is indeed included in the text somewhere.

*This is on Line 734: 'The stability metric introduced in this paper could in fact be a useful tool in future studies focusing on ocean-driven regime variability, or indeed in analysing non-stationary regime frameworks.'*

3. Original comment on ll. 68-69. What I meant to say here is that picking "things" that are well-represented to evaluate models and using the fact that they are well-represented as an argument that they are good evaluation metrics comes with the risk of a circular argument. I think the authors now make a strong point for why their regimes are useful in a model evaluation context, but in general they should be careful in using the lack of spatial regime variability (to use the terminology from Fig. 1) as an argument to support the usefulness of their regimes. One may indeed argue that, if there are robust regimes in the "real world" that are poorly spatially reproduced by the models, they can still provide useful model evaluation information, as long as they are not forced into a "temporal variability" framework with the spatial discrepancy being ignored.

*We appreciate both this and the original comment, which have helped us to frame and clarify the motivation and application of this work. We believe we now avoid any such circular reasoning in this draft, and more concretely detail the technical challenges posed by non-stationary regime patterns in the introduction, as well as the benefits of stable regimes in terms of clear relationships between model features and regime dynamics in section 5.*

4. L. 475 "Th"
*Corrected to 'The'.*

5. Original comment on l. 408. This is a minor, and mostly linguistic point, and I am perhaps being too picky. I follow the authors' argument, but I still do not understand how this says something about the "correctness" of individual models. If I understand correctly what the figure is showing, the standard deviation here combines information from different models, so at most it says something about CMIP5 or CMIP6 as a bulk as opposed to information about individual models. It also gives us no information about how many models fall within 1 standard deviation of the mean. Perhaps a more precise statement could be: "how the individual models differ in representing the correct level of…"

*The figure shows the standard deviation **in time**, averaged across the models, not the standard deviation **between** models. We have added this small clarification to the text to avoid confusion. As we believe this to be the source of the disagreement on wording, we stand by the current phrasing: whether CMIP5 or CMIP6 models have the 'correct amount' of variability or not is indeed shown by the difference between the model and reanalysis curves in figure 12.*

**Reviewer 2**

141 is 'non-gaussian' actually referred to the phase space?

*No, strictly speaking it refers to the distribution of points within the phase space, and so now we now write '...produces a phase space **distribution** which is unambiguously non-Gaussian…' to be more precise.*

170 I presume members of the ERA family share some similarities and so do members of the 20CR family. Also some are coupled some are not, maybe provide just a bit of text to comment on this.

*We now include a couple of sentences on this on line 171: 'Of these only CERA20C uses a coupled ocean-atmosphere model. Reanalyses produced by the same centre will share some similarities in the features of the assimilating model and in data-assimilation procedures, and so therefore are not totally independent'*

Figure 9, caption , after c) put a capital letter.
Figure 13 It appears that the figure is cropped at the bottom
Figure 13 and Figure 14, the labels are not explained, some are intuitive but only if one navigates through the paper carefully. 'Driv' is unclear. Please make sure that the labels are clearly defined. Also in figure 15 Stdev10 is not very rigorous

*The figure labels and captions have been clarified and corrected.*

First sentence of section 6, 'The two most important aspects…' you may refine the choice of words here
*Changed to 'two important aspects'.*